# Exosomes derived from HIV-1-infected cells promote growth and progression of cancer via HIV TAR RNA

Lechuang Chen[1], Zhimin Feng[1], Hong Yue[1,10], Douglas Bazdar[2], Uri Mbonye[3], Chad Zender[4,5], Clifford V. Harding [5,6,7], Leslie Bruggeman [7,8], Jonathan Karn [3,5,7], Scott F. Sieg [2,7], Bingcheng Wang[5,9] & Ge Jin [1,5,7]

People living with HIV/AIDS on antiretroviral therapy have increased risk of non-AIDS-defining cancers (NADCs). However, the underlying mechanism for development and progression of certain NADCs remains obscure. Here we show that exosomes released from HIV-infected T cells and those purified from blood of HIV-positive patients stimulate proliferation, migration and invasion of oral/oropharyngeal and lung cancer cells. The HIV transactivation response (TAR) element RNA in HIV-infected T-cell exosomes is responsible for promoting cancer cell proliferation and inducing expression of proto-oncogenes and Toll-like receptor 3 (TLR3)-inducible genes. These effects depend on the loop/bulge region of the molecule. HIV-infected T-cell exosomes rapidly enter recipient cells through epidermal growth factor receptor (EGFR) and stimulate ERK1/2 phosphorylation via the EGFR/TLR3 axis. Thus, our findings indicate that TAR RNA-containing exosomes from HIV-infected T cells promote growth and progression of particular NADCs through activation of the ERK cascade in an EGFR/TLR3-dependent manner.

[1] Department of Biological Sciences, Case Western Reserve University School of Dental Medicine, Cleveland, OH 44106, USA. [2] Department of Medicine, Division of Infectious Diseases and HIV Medicine, Case Western Reserve University School of Medicine, Cleveland, OH 44106, USA. [3] Department of Molecular Biology & Microbiology, Case Western Reserve University School of Medicine, Cleveland, OH 44106, USA. [4] Department of Otolaryngology/ENT Institute, Case Western Reserve University School of Medicine and University Hospitals Cleveland Medical Center, Cleveland, OH 44106, USA. [5] Case Comprehensive Cancer Center, Case Western Reserve University, Cleveland, OH 44106, USA. [6] Department of Pathology, Case Western Reserve University School of Medicine and University Hospitals Cleveland Medical Center, Cleveland, OH 44106, USA. [7] Center for AIDS Research, Case Western Reserve University and University Hospitals of Cleveland, Cleveland, OH 44106, USA. [8] Department of Inflammation and Immunity, Cleveland Clinic Lerner College of Medicine, Case Western Reserve University, Cleveland, OH 44195, USA. [9] Department of Medicine, Pharmacology and Oncology, Case Western Reserve University School of Medicine, Cleveland, OH 44106, USA. [10]Present address: Department of Biomedical Sciences, Joan C. Edwards School of Medicine, Marshall University, Huntington, WV 25701, USA. Correspondence and requests for materials should be addressed to G.J. (email: ge.jin@case.edu)

Cancer is a major cause of mortality and morbidity in AIDS patients and in chronically HIV-infected people. In the era of antiretroviral therapy (ART), the incidence of AIDS-defining cancers, such as Kaposi's sarcoma and several types of B-cell lymphomas, has been dramatically reduced[1]. However, non-AIDS-defining cancers (NADCs), such as head and neck squamous cell carcinoma (HNSCC) and lung cancers, have increased in HIV-infected people who are treated with ART mainly due to prolonged life span and aging[2,3]. Recent epidemiological studies indicate that cancer risk is elevated among older people living with HIV; the excess absolute risks have increased with age for lung, oral cavity/pharyngeal, anal, and liver cancers[4]. However, it remains unknown whether HIV-infected cells are involved in the development and progression of NADCs.

Most types of cells can release membrane-enclosed vesicles, generally called extracellular vesicles (EVs), into the extracellular space for intercellular communication, molecular transfer, and immune regulation at local and distant sites[5]. EVs are highly heterogeneous and dynamic and can be generally grouped into exosomes[6,7], macrovesicles[8], and apoptotic bodies based on biogenesis and the origin of vesicles[9]. Exosomes are generated as intraluminal vesicles that bud away from the cytoplasm into an intermediate endocytic compartment termed the multivesicular body (MVB) and then shed from cells upon fusion of MVB with the plasma membrane[7,10,11]. Exosomes contain various molecular cargoes of their cells of origin, including proteins and RNAs[11]. Although commonly used exosome purification protocols in the literature often co-isolate different types of EVs, the differential ultracentrifugation method isolates EVs that contain CD63, CD81, and CD9 tetraspanins and endosome marker-enriched vesicles which are characteristics of exosomes[11,12].

Exosomes can be isolated from culture media of HIV-1-infected cells and sera of people with HIV infection[13,14]. Latently HIV-1-infected Jurkat cell (J1.1) exosomes do not contain HIV-1 viral particles, although these exosomes contain viral proteins such as Gag and the precursor form of Env protein (p160)[13]. The HIV transactivation response (TAR) element RNA, a precursor of several HIV-encoded miRNAs, forms a stem–loop folding structure in the nascent transcript and facilitates binding of the viral transcriptional trans-activator (Tat) protein to enhance transcription initiation and elongation of HIV[15]. Exosomes isolated from HIV-1-infected cell culture supernatants or from HIV-infected patient sera contain TAR RNA in vast excess of total viral RNA[13]. TAR RNA-bearing exosomes significantly induce proinflammatory cytokines interleukin-6 (IL-6) and tumor necrosis factor-β (TNF-β) in primary macrophages[14].

Here, we report that exosomes derived from latently and actively HIV-1-infected T cells directly stimulate proliferation, migration, and invasion of HNSCC and lung cancer cells in vitro and promote tumor growth in xenograft animal models in vivo. Exosomes isolated from plasma of HIV-infected individuals under ART significantly promote cancer cell proliferation and migration compared with those from plasma of healthy people. However, exosome-depleted plasma from HIV-positive persons fails to enhance cancer cell proliferation. The HIV TAR RNA in HIV-infected T-cell exosomes is responsible for the pro-tumor effect and expression of the proto-oncogene FOS and TLR3-inducible interferon-stimulated genes (ISGs) in cancer cells, depending on the loop/bulge region of the molecule. HIV-infected T-cell exosomes quickly enter recipient cells via epidermal growth factor receptor (EGFR) and subsequently stimulate ERK1/2 (extracellular signal-regulated kinase 1 and 2) phosphorylation in HNSCC and lung cancer cells in an EGFR/Toll-like receptor 3 (TLR3)-dependent manner. Our data indicate that TAR RNA-bearing exosomes activate the ERK1/2 cascade in

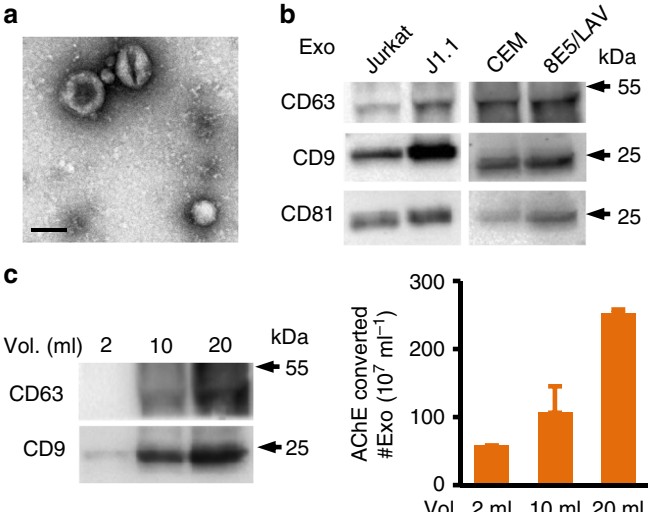

**Fig. 1** Characterization of exosomes from HIV-infected T cells. **a** Transmission electron microscope (TEM) images of exosomes isolated from T-cell culture supernatants. The representative J1.1 cell exosome image is shown. Scale bar, 100 nm. **b** Immunoblot of CD63, CD9, and CD81 on proteins extracted from T-cell line exosomes (whole scans of blots in Supplementary Figure 1). **c** CD63 and CD9 immunoblot (left) and AChE assays (right) of J1.1 exosomes isolated from 2, 10, and 20 ml of culture supernatants. Error bars, ± s.d. Data shown one experiment from three biological repeats. Numbers of exosomes (#Exo) were calculated by AChE activity using a standard provided by SBI

association with EGFR and TLR3 to promote proliferation, migration, and invasion of HNSCC and lung cancer cells.

## Results

**HIV-infected T-cell exosomes are cancer cell promoting.** J1.1[16], a latently HIV-1-infected human Jurkat T-cell line, 8E5/LAV[17], a LAV-infected CD4+ human CEM T-cell line, and their respective non-HIV control lines Jurkat and CEM were cultured in media supplemented with fetal bovine serum (FBS) that was exosome depleted by ultracentrifugation at $100,000 \times g$ for 16 h at 4 °C[18]. Exosomes were isolated from cell culture supernatants and human plasma using differential ultracentrifugation[14,18] and were in the size of 30–200 nm in diameter as determined by transmission electron microscope (Fig. 1a, Supplementary Fig. 1a, b). Members of the tetraspanin protein family CD9, CD63, and CD81, widely used as exosome markers[19], were detected in exosomes from all four cell lines by immunoblotting (Fig. 1b, Supplementary Fig. 1d to g for uncropped blots). The intensity of immunoblot images of CD63 and CD9 (Fig. 1c, left) was proportionally increased with incremental numbers of exosomes quantified using the EXOCET exosome quantification assay (System Bio. Inc., Palo Alto, CA), which converted the exosomal acetylcholinesterase (AChE) activity to numbers of exosomes (Fig. 1c, right)[20,21]. For each set of experiments in this report, we applied the same concentrations of exosomes in the range of 1 to $7.5 \times 10^9$ exosomes ml$^{-1}$ as indicated using the EXOCET assay which were equivalent to 30–180 μg ml$^{-1}$ of total exosomal proteins (Supplementary Fig. 1c).

To evaluate the effect of exosomes from HIV-positive or -negative T cells on cancer cells, we treated HNSCC, human lung cancer, and murine lung cancer cell lines with exosomes isolated from culture supernatants of HIV-1-infected J1.1 cells or from control Jurkat cells, respectively. J1.1 cell exosomes significantly promoted proliferation of HNSCC cells (Fig. 2a) as well as human H1299[22] and mouse K125 lung cancer cells (Supplementary

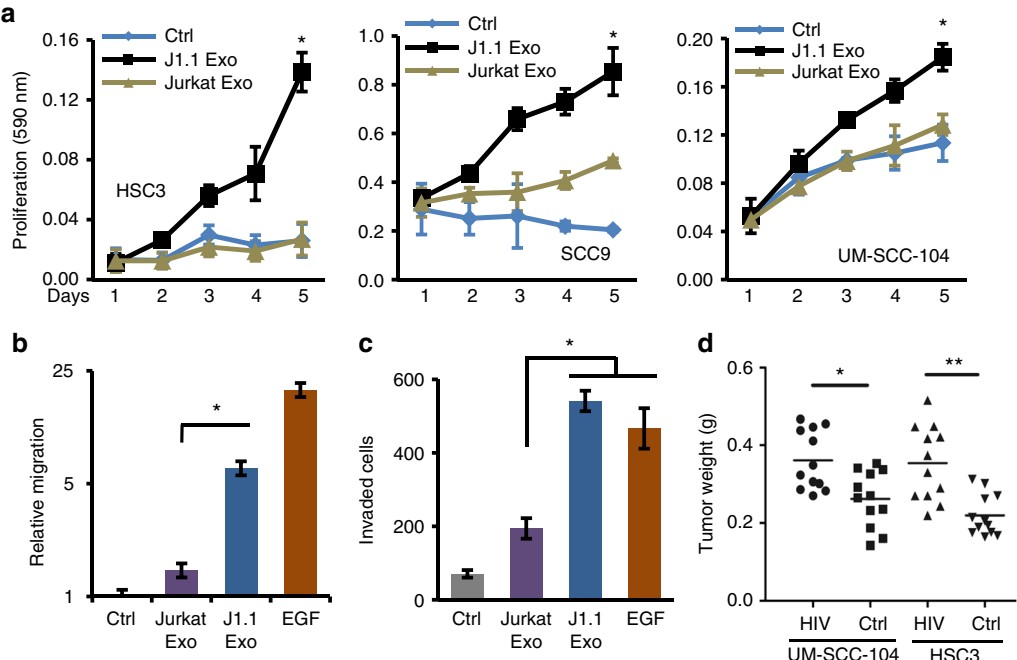

**Fig. 2** HIV-infected T-cell exosomes promote cancer cell proliferation and migration. **a** The effect of exosomes on cell proliferation of HSC3, SCC9, and UM-SCC-104 HNSCC cells treated with $4 \times 10^9$ exosomes $ml^{-1}$ from J1.1 (J1.1 Exo) or Jurkat (Jurkat Exo) cells. Ctrl, serum-free medium. Error bars, ± s.d., $n = 3$, $*p < 0.01$, one-way ANOVA on days 3 to 5. **b** HSC3 cancer cell migration in response to J1.1 (J1.1 Exo) or Jurkat cell exosomes (Jurkat Exo) ($4.2 \times 10^9$ $ml^{-1}$). Error bars, ± s.d., $n = 4$, $*p < 0.02$, $F$-test. EGF (10 ng $ml^{-1}$), positive control. Relative migration was determined by migration of treatment over that of control. **c** Transwell invasion assays of HSC3 cells in response to J1.1 (J1.1 Exo) or Jurkat cell exosomes (Jurkat Exo) ($4.2 \times 10^9$ $ml^{-1}$) loaded in the lower chamber of the transwell plate. EGF (10 ng $ml^{-1}$), positive control. Error bars, ± s.d., $n = 4$, $*p < 0.02$, $F$-test. **d** Tumor weight of UM-SCC-104 and HSC3 cell xenografts in nude mice. Cancer cells ($5 \times 10^6$) were mixed with $1.6 \times 10^{10}$ $ml^{-1}$ of J1.1 or Jurkat cell exosomes and then inoculated into nude mice for the xenograft experiment. Data represent means ± s.d., $n = 12$, $*p < 0.05$ and $**p < 0.02$, Student's $t$-test. Data represent one experiment from at least three biological repeats in **a**, **b** and **c**

Fig. 2a, b) compared with control Jurkat cell exosome treatment. Dose–response results indicated that HSC3 cancer cells proliferated proportionally with higher concentrations of J1.1 cell exosomes, in the range of $2-8 \times 10^9$ exosomes $ml^{-1}$ (Supplementary Fig. 2c). Exosomes from J1.1 cells, but not those from Jurkat cells, significantly increased HNSCC HSC3 cell migration and invasion through the Matrigel matrix (Fig. 2b, c). J1.1 cell exosomes also promoted migration of H1299 human lung cancer cells (Supplementary Fig. 2d). These results indicate that exosomes released from HIV-infected T cells are able to promote malignant behavior of cancer cells in vitro.

To evaluate the effect of HIV-1-infected T-cell exosomes on tumorigenesis in vivo, HNSCC and lung cancer cells were inoculated into athymic nude mice in the presence of exosomes from HIV-infected J1.1 cells or those from control Jurkat cells. Xenograft tumors of HNSCC cells inoculated together with J1.1 cell exosomes grew faster and exhibited significantly higher tumor weight at the end of the experiment compared with those treated with Jurkat exosomes (Fig. 2d). Similarly, J1.1 cell exosomes enhanced growth of xenograft tumors inoculated with H1299 lung cancer cells (Supplementary Fig. 2e). These results suggest that HIV-infected T-cell exosomes promote engraftment and growth of established cancer cells in vivo.

Since exosomes from cancer cells might signal T cells to cause reactivation of HIV, thus indirectly contributing to tumorigenesis, we tested the possibility of such reciprocal exosome-mediated signaling between cancer cells and HIV-infected T cells. Exosomes isolated from culture supernatants of HNSCC cell lines (Supplementary Fig. 3a) were incubated with the latently infected 2D10 Jurkat T cells which carry a green fluorescent protein (GFP) reporter in the provirus; therefore, GFP-positive 2D10 cells

indicate HIV-1 transactivation[23]. While TNFα (5 ng $ml^{-1}$) or phorbol 12-myristate 13-acetate (PMA, 1 nM) induced reactivation of HIV in 2D10 cells (Supplementary Fig. 3b)[23], cancer cell exosomes failed to stimulate transactivation of the HIV genome (Supplementary Fig. 3b, c), suggesting the unidirectional exosome signaling from HIV-infected T cells to cancer cells.

**Pro-tumor effect of HIV-infected T-cell exosomes needs HIV TAR RNA.** We proposed that HIV-specific exosomal cargo components participated in promoting cancer cell proliferation, migration, and invasion. To test this hypothesis, HSC3 cancer cells were treated with exosomes isolated from cell culture supernatants of latent J1.1 cells and those of HIV-positive 2D10 cells that lacked the viral *nef* gene[23]. Exosomes from both cell lines promoted proliferation of HSC3 cells at the same level (Fig. 3a). We then performed a cell migration assay using HSC3 cells in response to exosomes isolated from 2D10 cells and those from HIV-1-infected C22G T cells that contain a disruptive HIV *tat* mutant and *nef* deletion[24]. Exosomes from J1.1, 2D10 and C22G cells significantly stimulated HSC3 cancer cell migration at similar levels (Fig. 3b). Therefore, the pro-tumor effect of HIV-positive T-cell exosomes is independent of viral *tat* or *nef* transcript/protein. In addition, exosomes purified from latent or TNFα-activated J1.1 cells enhanced HSC3 cancer cell migration at the same level (Fig. 3b, J1.1 vs. J1.1+TNF), suggesting that the pro-tumor effect of HIV-positive T-cell exosomes is independent of HIV reactivation.

It has been reported that exosomes from HIV-1-infected T-cell lines, including J1.1, contain the HIV TAR element RNA in vast excess of any other viral RNAs, such as Tat, Env, and Nef[13,14]. Similarly, we found that exosomes from J1.1 cells contained

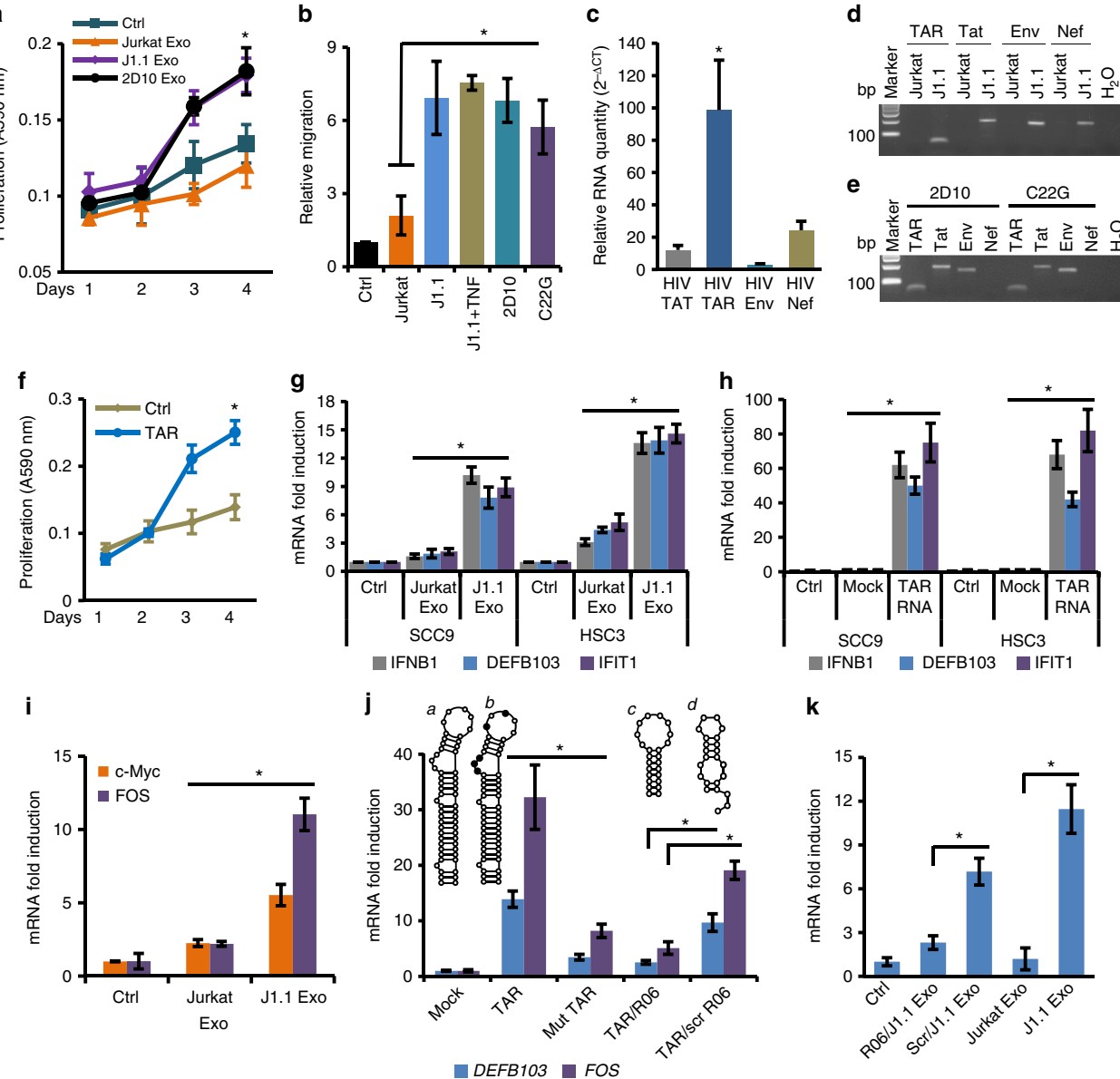

**Fig. 3** TAR RNA contributes to the pro-tumor function of HIV-positive exosomes. **a** HSC3 cells treated with J1.1, 2D10 and Jurkat cell exosomes ($4.1 \times 10^9$ ml$^{-1}$) for cell proliferation. Error bars, ± s.d., $n = 3$, *$p < 0.05$, one-way ANOVA (days 3 and 4). **b** HSC3 cells treated with J1.1, TNFα-activated J1.1 (J1.1+TNF), 2D10, C22G, and Jurkat cell exosomes ($4 \times 10^9$ ml$^{-1}$) for cell migration. Ctrl, serum-free control. Error bars, ± s.d., $n = 3$, *$p < 0.05$, F-test. **c** qRT-PCR of HIV RNA in J1.1 cell exosomes. Error bars, ± s.d., $n = 3$, *$p < 0.02$, F-test. **d** PCR amplimers in **c**. bp, DNA marker size. **e** RT-PCR of 2D10 and C22G cell exosomes. H$_2$O, mock PCR. **f** Proliferation of HSC3 cells transfected with synthetic TAR RNA (1 µg ml$^{-1}$). Error bars, ± s.d., $n = 3$, *$p < 0.05$, one-way ANOVA (days 3 and 4). Ctrl mock transfection. **g** qRT-PCR of *IFNB1*, *DEFB103*, and *IFIT1* mRNA on cells treated with Jurkat and J1.1 cell exosomes ($4 \times 10^9$ ml$^{-1}$) for 18 h. Exo, exosomes. Error bars, ± s.d., $n = 3$, *$p < 0.05$, F-test. **h** qRT-PCR of SCC9 and HSC3 cells transfected with synthetic TAR RNA (1 µg ml$^{-1}$) for 18 h. Error bars, ± s.d., $n = 3$, *$p < 0.05$, F-test. Mock, mock transfection; ctrl, un-transfected. **i** qRT-PCR for c-Myc and *FOS* mRNA in HSC3 cells treated with exosomes ($4 \times 10^9$ ml$^{-1}$) for 18 h. Error bars, ± s.d., $n = 3$, *$p < 0.05$, F-test. **j** qRT-PCR of *FOS* and *DEFB103* mRNA in HSC3 cells transfected with TAR RNA (TAR, *a*), mutant TAR (Mut TAR, *b*), co-transfected with R06 aptamer (TAR/R06; *c*) or scrambled RNA (TAR/scr R06; *d*) for 18 h. Error bars, ± s.d., $n = 3$, *$p < 0.05$, F-test. **k** qRT-PCR of *DEFB103* in HSC3 cells treated with J1.1 exosomes transfected with R06 or scrambled aptamer (Scr) for 18 h. Error bars, ± s.d., $n = 3$, *$p < 0.05$, F-test. Jurkat and J1.1 cell exosome ($4 \times 10^9$ ml$^{-1}$) treatment were used as controls. Data in **a**, **b**, **c**, **f**, **g**, **h**, **i**, **j** and **k**) represent one experiment from three independent repeats

significantly higher levels of TAR RNA compared to Tat, Env, and Nef RNA by quantitative reverse transcription-polymerase chain reaction (qRT-PCR) (Fig. 3c). The qRT-PCR amplimers showed predicted sizes for each HIV RNA (Fig. 3d). The TAR RNA, together with Tat and Env RNA albeit without Nef expression, was also identified in exosomes isolated from culture media of 2D10 and C22G cells (Fig. 3e). Therefore, we postulated that the TAR RNA in HIV-infected T-cell exosomes was

responsible for promoting cancer cell proliferation. To test this hypothesis, we transfected HSC3 cells with the synthetic TAR RNA and found that the TAR RNA alone strongly promoted HSC3 cell proliferation (Fig. 3f), suggesting that the TAR RNA was critical for the pro-tumor effects of exosomes derived from HIV-infected T cells.

It has been reported that TAR RNA-bearing exosomes from HIV-1-infected J1.1 cells induce production of proinflammatory

cytokines IL-6 and TNF-β in primary monocyte-derived macro-phages[14]. We found that exosomes from J1.1 cells stimulated significant expression of TLR3-responsive, ISGs *IFIT1* and *IFNB1*[14,25,26] as well as an EGF-inducible *DEFB103* which encodes human β-defensin-3 (hBD-3) that is produced by proliferating oral cancer cells and involved in macrophage trafficking in tumors[27] (Fig. 3g). To determine whether the TAR RNA was able to directly induce expression of the same genes, we transfected TAR RNA into HSC3 and SCC9 oral cancer cells and found that TAR RNA significantly increased *IFIT1*, *IFNB1*, and *DEFB103* expression in the cancer cells (Fig. 3h). Therefore, our data indicate that the synthetic HIV TAR RNA can directly stimulate expression of the same genes that are induced by exosomes from HIV-infected T cells.

The tumor-promoting function of HIV-infected T-cell exo-somes suggests that the exosomes are able to activate genes of oncogenic network of cancer cells. We investigated if exosomes from HIV-infected T cells would induce expression of oncogenes in cancer cells. Our results showed that J1.1 T-cell exosomes stimulated significant expression of proto-oncogenes *FOS*[28,29] and, to a lesser extent, c-Myc mRNA[30,31] compared to Jurkat cell exosomes (Fig. 3i), suggesting that exosomes from HIV-infected T cells can directly promote cell proliferation through up-regulation of proto-oncogenes in HSC3 cancer cells.

The HIV TAR RNA forms a stable stem–bulge–loop structure[32]. While the bulge–loop region is required for HIV Tat protein binding[33], the stem structure has been described as a double-stranded RNA (dsRNA) binding domain[34]. To determine whether the bulge–loop or the dsRNA-like stem of TAR RNA played a role in gene expression, we transfected HSC3 cells with a TAR RNA mutant containing 5-nucleotide replacements in bulge and loop sequences. This TAR RNA mutant prevents Tat binding but remains the basic stem–bulge–loop structure[35] (Fig. 3j; inset *a*, wild-type TAR RNA; *b*, mutant TAR RNA, nucleotide replacements designated in solid circulars). While wild-type TAR RNA induced expression of *FOS* and *DEFB103* in HSC3 cells, the mutant TAR RNA failed to stimulate expression of the genes (Fig. 3j, TAR vs. Mut TAR). Furthermore, the RNA aptamer R06 (Fig. 3j, inset *c*), which is complementary to the TAR apical region and blocks TAR function without disrupting the secondary structure of TAR[36], inhibited TAR RNA-induced *FOS* and *DEFB103* expression in HSC3 cells (Fig. 3j, TAR/R06). However, a scrambled aptamer (Fig. 3j, *d*) was unable to inhibit TAR RNA-induced gene expression (Fig. 3j, TAR/scr R06). To verify that J1.1 exosome-induced *DEFB103* was TAR RNA dependent, J1.1 cell exosomes were transfected with the R06 aptamer or its scrambled counterpart, followed by stimulating HSC3 cells with the exosomes for *DEFB103* expression. J1.1 cell exosomes transfected with the R06 aptamer were unable to induce significant *DEFB103* expression in HSC3 cells, while those transfected with scrambled aptamer still stimulated expression of *DEFB103* (Fig. 3k, Scr/J1.1 Exo vs. R06/J1.1 Exo). These results indicate that the bulge–loop region of TAR RNA is critical for the pro-tumor function of TAR RNA-bearing exosomes.

**Plasma of HIV-infected persons contains pro-tumor exosomes**. Since latently HIV-infected T-cell exosomes promoted cancer cell proliferation, migration, and invasion, we proposed that chroni-cally HIV-infected people who were under ART would contain tumor-promoting TAR RNA-bearing exosomes. To test this hypothesis, we treated cancer cells with exosomes isolated from plasma specimens of six HIV-positive patients and six HIV-negative control individuals for cancer cell proliferation assays. All HIV-positive subjects had CD4+ T-cell counts over 200 per ml and were under ART treatment (Supplementary Table 1).

Plasma exosomes from HIV-infected subjects significantly induced proliferation of HSC3 HNSCC and H1299 lung cancer cells (Fig. 4a). Importantly, exosomes purified from plasma of HIV-positive HNSCC patients (Supplementary Table 2) stimu-lated HSC3 cancer cell proliferation (Fig. 4b). Interestingly, exo-somes isolated from sera of HIV-transgenic Tg26 mice (HIV+), which contained a defective HIV-1 proviral genome controlled by the native viral promoter (long terminal repeat, LTR)[37,38], but not those from sera of syngeneic wild-type FVB strain littermates (HIV−), significantly induced human lung cancer cell migration (Fig. 4c).

Our results suggested that HIV-positive subjects contained pro-tumor exosomes in their circulation. To determine whether the concentration of exosomes in the HIV-positive patient plasma was high enough to yield the pro-tumor effect shown in Fig. 4a, b, we treated HSC3 cells with unmodified primary plasma, purified plasma exosomes in serum-free Dulbecco's modified Eagle's medium (DMEM), and resulting exosome-depleted plasma from the same HIV-positive and HIV-negative subjects, followed by cell proliferation assays. Plasma exosomes purified from HIV-positive patients, but not those from healthy individuals, stimulated HSC3 cell proliferation (Fig. 4d, left). Interestingly, the primary plasma from HIV-positive patients significantly promoted cancer cell proliferation compared with that from healthy subjects (Fig. 4d, middle). Exosome-depleted plasma (exo-depl plasma) from the same HIV-positive patients, however, failed to stimulate HSC3 cell proliferation relative to that from healthy control people (Fig. 4d, right). Purified plasma exosomes induced higher levels of cell proliferation than the unmodified plasma containing the same amount of exosomes from HIV-positive patients (average A590, 0.8 for exosomes vs. 0.6 for primary plasma). To assess the concentration of exosomes in the plasma of HIV-positive and HIV-negative individuals, we quantified exosomes in unmodified plasma, exosomes purified from the same plasma samples, and the resulting exosome-depleted plasma using the AChE assay. Our results showed that the exosome-depleted plasma contained negligible amount of exosomes (Supplementary Table 3). Although the human plasma comprises not only exosomes but also other types of EVs and soluble protein complexes, our results suggest that the HIV-positive patient's plasma contributes to cancer cell proliferation mainly through circulating exosomes.

To determine whether plasma exosomes from HIV-positive people contained the HIV TAR RNA, qRT-PCR was performed using total exosomal RNA from HIV-infected subjects and HIV-positive HNSCC patients. Exosomes from all six HIV-positive subjects, but not those from non-HIV control individuals, contained TAR RNA which was 7- to 25-fold higher over that in exosomes from HIV-infected 8E5/LAV cells which contain a single integrated copy of proviral HIV DNA (Fig. 4e). Exosomes from five HIV-positive HNSCC patient blood contained TAR RNA that was 20- to 80-fold higher than that in 8E5/LAV cell exosomes (Fig. 4f; bottom, TAR RNA RT-PCR amplimer on agarose gel). RNA extracted from formalin-fixed paraffin-embedded (FFPE) sections of HIV-positive HNSCC patient biopsies, but not from non-HIV HNSCC patients, contained HIV TAR RNA (Fig. 4g; bottom, RT-PCR amplimer on agarose gel). In addition, TAR RNA was detected in blood of Tg26 HIV-transgene mice (Fig. 4h). Our results indicate that HIV-positive people contain circulating pro-tumor HIV TAR RNA-bearing exosomes and TAR RNA-containing cells/exosomes in the tumor niche of NADCs.

**HIV-infected T-cell exosomes stimulate the EGFR–ERK sig-naling**. Activation of the ERK mitogen-activated protein kinase (MAPK) cascade functions in growth, progression, and survival

of human cancers and is linked with EGFR which is involved in HNSCC and lung cancer progression[39,40]. We evaluated the role of the ERK cascade in cancer cell proliferation, migration, and invasion in response to J1.1 cell exosomes. Treatment of cancer cells with U0126, a selective MAPK/ERK kinase1/2 (MEK1/2) inhibitor that blocks ERK1/2 phosphorylation[41], or with cetuximab, a humanized monoclonal antibody to EGFR[42], inhibited J1.1 cell exosome-induced HSC3 cancer cell proliferation (Fig. 5a)

and invasion (Fig. 5b), respectively. To determine the role of HIV-positive exosomes in ERK signaling, we treated HSC3 HNSCC and H1299 lung cancer cells with exosomes purified from J1.1 or Jurkat cells, followed by immunoblotting for ERK1/2 activation. Exosomes from J1.1 cells specifically stimulated phosphorylation of ERK1/2 in both HSC3 and H1299 cancer cells, which was blocked by cetuximab (Fig. 5c). However, J1.1 cell exosomes failed to induce canonical phosphorylation of

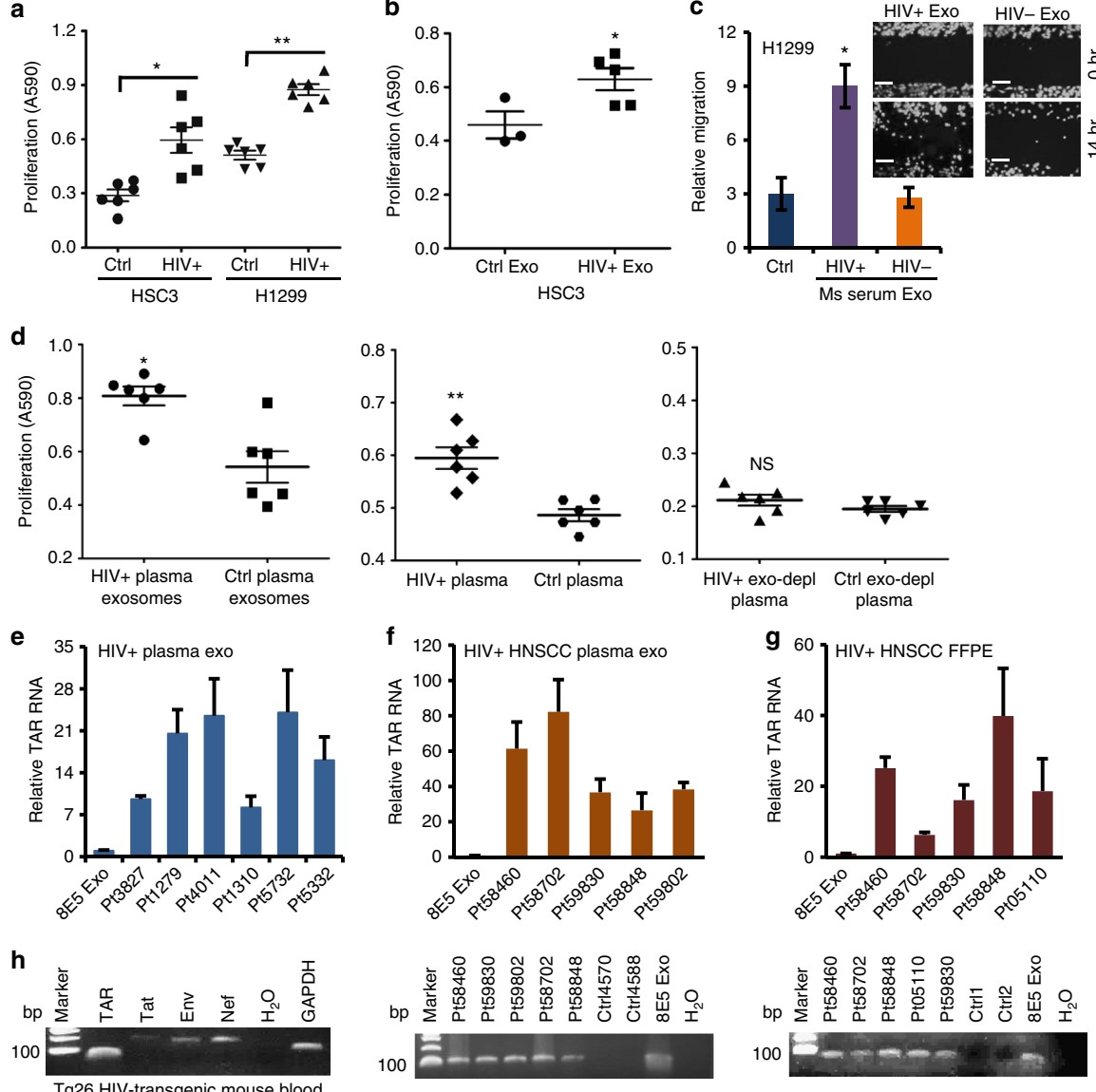

**Fig. 4** Plasma exosomes from HIV-infected patients promote cancer cell growth. **a** Proliferation of HSC3 and H1299 cells treated with exosomes ($3.9$–$4.5 \times 10^9$ ml$^{-1}$) from the plasma of normal (Ctrl) and HIV-infected subjects (HIV+) for 5 days. Error bars, ± s.d., $n = 6$; *$p < 0.002$ and **$p < 0.0001$, Student's $t$-test. **b** Proliferation of HSC3 cells treated with exosomes ($4.1$–$4.8 \times 10^9$ ml$^{-1}$) from plasma of normal (Ctrl, $n = 3$) and HIV-positive HNSCC subjects (HIV +Exo, $n = 5$) for 5 days. Error bars, ± s.d., *$p < 0.04$, one-way ANOVA. **c** Migration of H1299 lung cancer cells treated with exosomes from sera of Tg26 mice (HIV+) or FVB control littermates (HIV−) ($4.2 \times 10^9$ ml$^{-1}$). Ms, mouse; Exo, exosomes. Error bars, ± s.d., $n = 3$, *$p < 0.05$, $F$-test. Data of one experiment from two repeats are shown. Inset, migration images; scale bars, 100 μm. **d** Proliferation of HSC3 cells treated with the unmodified plasma ($4.7 \times 10^9$ exosomes ml$^{-1}$, left), purified plasma exosomes ($4.7 \times 10^9$ ml$^{-1}$, middle), and exosome-depleted plasma (right, exo-depl) from HIV-positive (HIV+) or healthy people (Ctrl) on day 5. Error bars, ± s.d., $n = 6$; * and **$p < 0.05$, Student's $t$-test. NS, $p > 0.05$. **e** qRT-PCR of TAR RNA on total RNA of plasma exosomes from HIV-positive subjects (HIV+ plasma exo). Relative TAR RNA levels were compared to TAR RNA of 8E5/LV cell exosomes. Error bars, ± s.d. $n = 3$. 8E5 Exo, 8E5/LV cell exosomes. **f** qRT-PCR of TAR RNA of plasma exosomes from HIV-positive HNSCC patients (top). Relative TAR RNA expression is determined as in **e**. Errors bars, ± s.d., $n = 3$. Bottom, TAR RNA gel image; Ctrl4570 and Ctrl4588, exosomes from HIV-negative HNSCC patient plasma. **g** qRT-PCR of TAR RNA using total RNA extracted from FFPE sections of HIV-positive HNSCC patients (top). Relative TAR RNA levels were determined as in **e**. Error bars, ± s.d., $n = 3$. Bottom, TAR RNA gel image; Ctrl1 and Ctrl2, HIV-negative HNSCC FFPE sections. **h** RT-PCR of HIV RNA in whole blood RNA from Tg26 HIV-transgenic mice

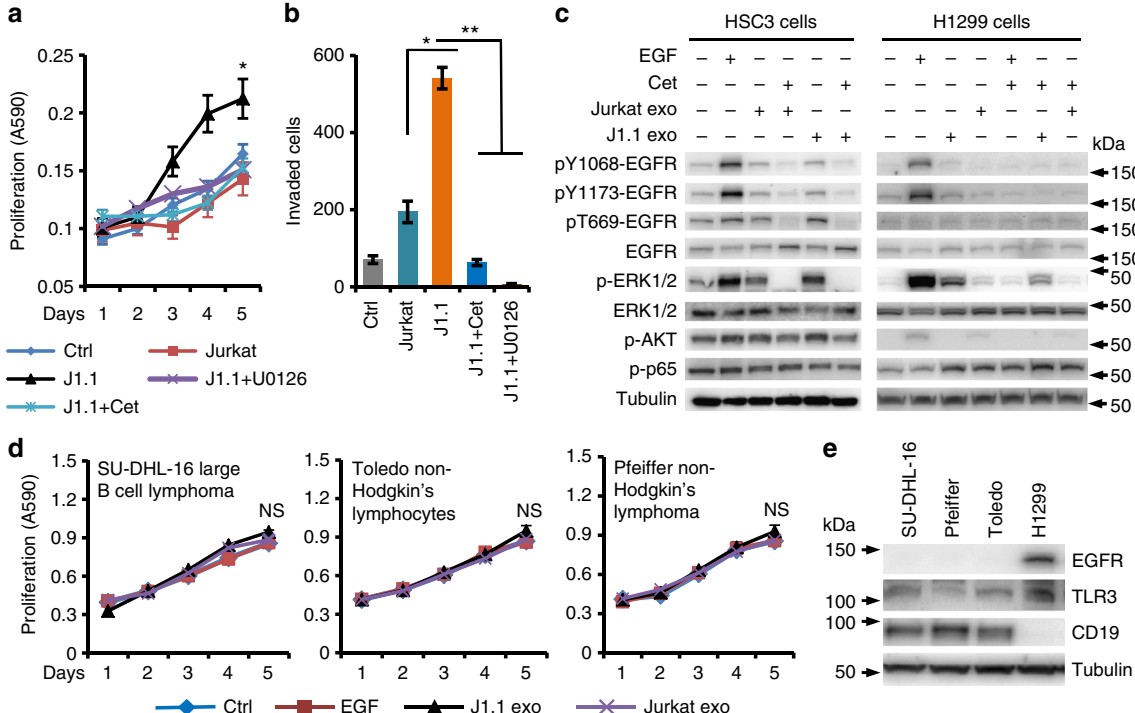

**Fig. 5** J1.1 cell exosomes stimulate ERK1/2 phosphorylation via EGFR. **a** HSC3 cells were treated with U0126 (5 μM, + U0126) or cetuximab (20 μg ml$^{-1}$, +Cet) for 30 min or remained untreated, followed by stimulation with exosomes ($4 \times 10^9$ ml$^{-1}$) from J1.1 or Jurkat cells for cell proliferation. Ctrl, serum-free medium; J1.1 and Jurkat, exosomes from J1.1 and Jurkat cells, respectively. Error bars, ± s.d., $n = 4$, *$p < 0.05$, one-way ANOVA on days 3 to 5. **b** HSC3 cell invasion in response to exosomes ($4.2 \times 10^9$ ml$^{-1}$) from Jurkat or J1.1 cells in the absence or presence of U0126 (5 μM, +U0126) or cetuximab (20 μg ml$^{-1}$, +Cet). Data (mean ± s.d., $n = 3$) represent one experiment out of three independent repeats; *$p < 0.05$ and **$p < 0.02$, $F$-test. **c** HSC3 HNSCC and H1299 lung cancer cells treated with exosomes ($2 \times 10^9$ ml$^{-1}$) from Jurkat or J1.1 cells in the absence or presence of cetuximab (20 μg ml$^{-1}$, Cet) for 10 min, followed by immunoblotting on total cell lysates. EGF (10 ng ml$^{-1}$), positive control; kDa, protein marker size. Uncropped blot images in Supplementary Figure 4e. **d** Proliferation of B-cell lymphoma cell lines SU-DHL-16, Toledo, and Pfeiffer in response to exosomes ($4 \times 10^9$ ml$^{-1}$) isolated from J1.1 or Jurkat cells and EGF (10 ng ml$^{-1}$). Data (mean ± s.d., $n = 4$) represent one experiment from two independent repeats. NS, $p > 0.05$, one-way ANOVA on days 3 to 5. **e** Immunoblot of total EGFR (EGFR), TLR3, and CD19 using total cell lysates from SU-DHL-16, Pfeiffer, Toledo, and H1299 cells. Uncropped blot images in Supplementary Figure 4f

EGFR at tyrosine residuals for receptor activation (Fig. 5c, pY1068-EGFR and pY1173-EGFR. pY, phosphor-tyrosine). Exosomes from control Jurkat cells only moderately caused ERK1/2 phosphorylation in HSC3 cells (Fig. 5c). EGF treatment was used as a positive control and, as expected, stimulated phosphorylation of EGFR, ERK1/2, and AKT (Fig. 5c). In addition, J1.1 cell exosomes induced phosphorylation of Erk1/2 in mouse Lewis lung cancer (LLC1) cells[43] (Supplementary Fig. 4a). Time-course analysis showed that HIV-infected T-cell exosomes induced ERK1/2 phosphorylation occurred as early as 10 min after treatment (Supplementary Fig. 4b, c). Our data suggest that J1.1 cell exosome-induced ERK1/2 phosphorylation is EGFR dependent without affecting canonical EGFR phosphorylation.

The involvement of EGFR in HIV-positive exosome signaling prompted us to propose that exosomes could enter recipient cells through EGFR. Indeed, cetuximab blocked entry of red fluorescently labeled J1.1 cell exosomes into HEK293 cells expressing GFP-tagged EGFR within about 10 min using the time-delayed $z$-series scanning confocal microscopy (Supplementary Movie 1). However, with incubation of exosomes with HSC3 cells for 9 to 18 h, all exosomes were inside of the cells (Supplementary Fig. 5a, b). Our findings indicated that EGFR mediated a quick entry of exosomes (~10 min), a time frame correlated with ERK1/2 phosphorylation induced by HIV-positive exosomes.

Lymphoma is a common type of AIDS-defining malignancy often diagnosed in people with AIDS[44]. To evaluate the effect of HIV-positive T-cell exosomes on B lymphoma cells, we treated the large B-cell lymphoma cell line SU-DHL-16 and two B-cell non-Hodgkin lymphoma lines Pfeiffer and Toledo with J1.1 or Jurkat cell exosomes. Interestingly, J1.1 cell exosomes did not affect proliferation of B lymphoma cells compared with Jurkat cell exosomes (Fig. 5d). We proposed that the unresponsiveness of B lymphoma cells to the pro-tumor effect of HIV-positive T-cell exosomes was due to the lack of EGFR in the cells. Indeed, all three B-cell lymphoma cell lines lacked EGFR expression, while they all expressed the B-cell marker CD19 (Fig. 5e). In contrast, the H1299 lung cancer cells strongly expressed EGFR (Fig. 5e). These results suggest that the EGFR/ERK signaling induced by HIV-positive exosomes is cancer type specific.

**TLR3 mediates TAR RNA-induced gene expression and signaling.** HIV-infected J1.1 cell exosomes and synthetic HIV TAR RNA induced significant expression of TLR3-responsive ISGs *IFIT1* and *IFNB1* in SCC9 and HSC3 cells (Fig. 3f, g), suggesting that TAR RNA-bearing exosomes from HIV-1-infected T cells would signal through TLR3 in cancer cells. Indeed, the TLR3/dsRNA inhibitor complex, a competitive and high affinity inhibitor of dsRNA binding to TLR3[45], blocked the capacity of J1.1 cell exosomes to induce expression of *IFIT1* and *IFNB1* in HSC3 cancer cells (Fig. 6a, J1.1+inh). Furthermore, the TLR3/dsRNA inhibitor reduced ERK1/2 phosphorylation in HSC3 cancer cells treated with J1.1 cell exosomes (Fig. 6b). To ultimately define the role of TLR3 in mediating HIV-positive exosome-induced cell

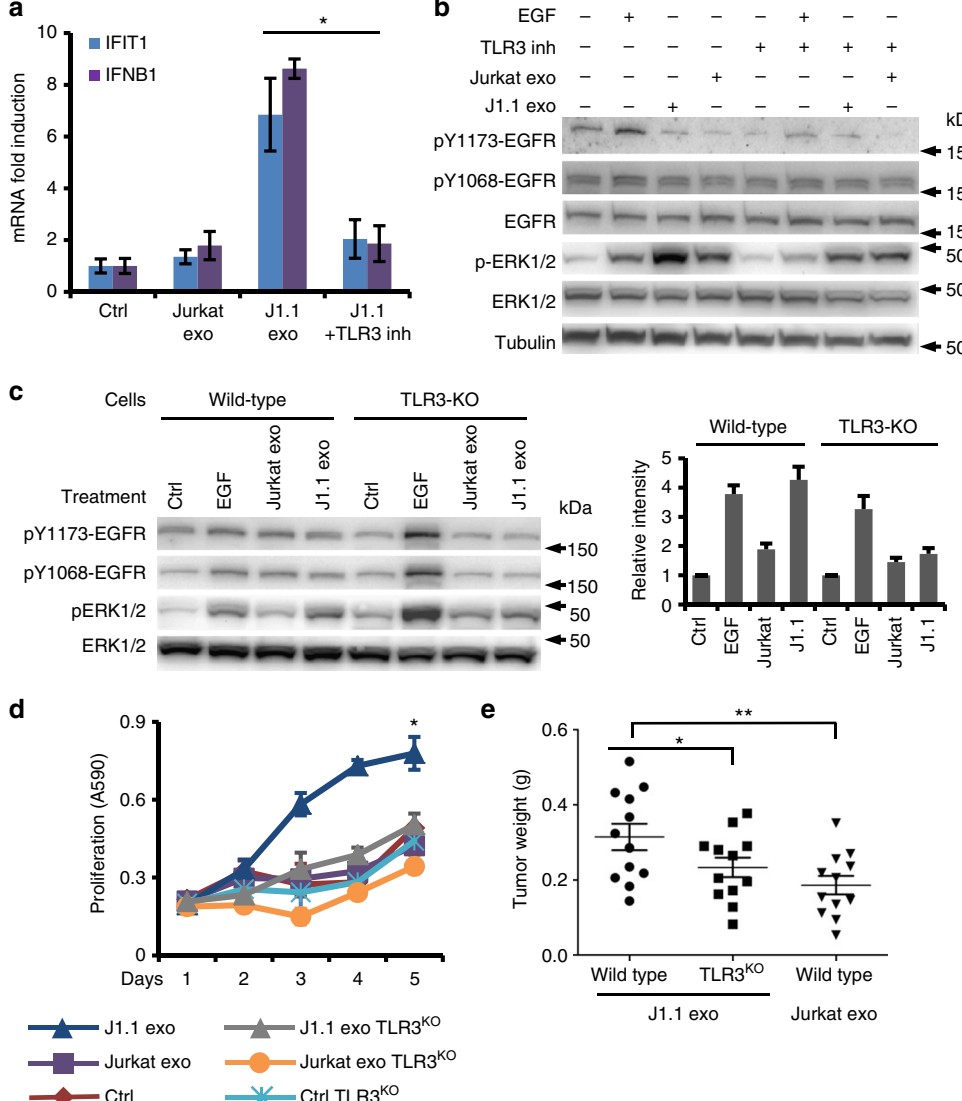

**Fig. 6** TLR3 is involved in tumor-promoting function of HIV-positive exosomes. **a** HSC3 cells pretreated with the TLR3/dsRNA inhibitor (6 μM, +Inh) for 30 min, or remained untreated, and then transfected with TAR RNA (1 μg ml$^{-1}$). Total RNA was used for qRT-PCR of *IFIT1* and *IFNB1* mRNA. Error bars, ± s. d., $n = 3$, *$p < 0.02$, *F*-test. Data represent one experiment from three independent repeats. **b** Immunoblot of HSC3 cancer cells treated with exosomes (2 × 10$^9$ ml$^{-1}$) from J1.1 or Jurkat cells in the presence or absence of the TLR3/dsRNA inhibitor (6 μM, TLR3 inh). Uncropped blot images in Supplementary Figure 4g. **c** Immunoblot of wild-type or TLR3-KO HSC3 cells treated with J1.1 or Jurkat cell exosomes at 2 × 10$^9$ ml$^{-1}$. Right, densitometry of immunoblots over that of control. Error bars, ± s.d. Uncropped blot images in Supplementary Figure 4h. **d** Wild-type (Ctrl) and TLR3-KO (TLR3$^{KO}$) HSC3 cancer cells were treated with exosomes (4 × 10$^9$ ml$^{-1}$) from J1.1 or Jurkat cells for proliferation assays. Data (mean ± s.d., $n = 3$) represent one experiment from three independent repeats; *$p < 0.05$, one-way ANOVA on days 3 to 5. **e** Tumor growth of wild type and TLR3-KO HSC3 (TLR3$^{KO}$) cancer cells inoculated into nude mice in the presence of J1.1 exosomes (J1.1 exo, 6.3 × 10$^9$ ml$^{-1}$). HSC3 cells inoculated with Jurkat exosomes (Jurkat exo, 6.3 × 10$^9$ ml$^{-1}$) were used as control. Error bars represent ± s.d., $n = 12$, *$p < 0.01$ and **$p < 0.007$, Student's *t*-test

signaling and proliferation, we generated TLR3-knockout (TLR3-KO) HSC3 cells using the CRISPR/double nickase (Supplementary Fig. 4d) and treated TLR3-KO cells with J1.1 and Jurkat exosomes for signaling and cell proliferation. While J1.1 cell exosomes considerably induced ERK1/2 phosphorylation in wild-type HSC3 cells, they did not cause significant ERK1/2 phosphorylation in TLR3-KO HSC3 cells (Fig. 6c; right, densitometry of immunoblots). Further, HIV-positive J1.1 cell exosomes were unable to enhance proliferation of TLR3-KO HSC3 cancer cells in vitro (Fig. 6d) and xenograft tumor growth in nude mice in vivo (Fig. 6e). TLR3-KO HSC3 cells did not show any growth disadvantage over wild-type cells (Fig. 6d, Ctrl TLR3$^{KO}$ vs. Ctrl). Because the TLR3-KO cells retained EGFR expression, our results

suggest that TAR RNA-bearing exosomes activates the ERK cascade through EGFR and TLR3.

Epstein–Barr virus (EBV) is a human herpesvirus and an oncogenic agent for several cancers, including nasopharyngeal cancer, Hodgkin's lymphoma, and Burkitt's lymphoma[46]. EBV-infected cells release exosomes that contain the EBV-encoded small RNAs (EBER1 and EBER2). The noncoding RNA forms stem–loop structure assembling dsRNA-like molecules and binds to TLR3 to induce expression of ISGs[47,48]. We identified EBER1, but not EBER2, in exosomes isolated from culture supernatants of EBV-infected human lymphoblast line NC-37[49] and the latently EBV-infected mouse lymphoblastoid cell line IB4[48] (Supplementary Fig. 6a). However, EBER1 and EBER2 were not detected in

exosomes derived from non-EBV B-cell lymphoma cell lines SU-DHL-16 and Toledo (Supplementary Fig. 6a). To determine the effect of EBV-infected lymphoblast exosomes on HNSCC and lung cancer cells, we treated HSC3 and H1299 cells with exosomes from culture supernatants of EBV+ and EBV− cells, followed by cell proliferation assays. Our results revealed that EBV+ and EVB− cell exosomes failed to induce proliferation of H1299 cancer cells, while EBV− cell exosomes only induced moderate proliferation of HSC3 cells on day 5 (Supplementary Fig. 6b, c). We also found that exosomes from EBV+ and EBV− cell lines failed to stimulate expression of *FOS*, c-Myc, and *IFNB1*, although the exosomes induced significant expression of the ISG *IFIT1* in HSC3 cancer cells (Supplementary Fig. 6d). Our results suggest that the pro-tumor effect on HNSCC and lung cancer cells was HIV-positive T-cell exosome specific.

## Discussion

Exosomes derived from HIV-1-infected T cells are involved in various disease processes of HIV infection, including neurotoxicity[50], production of proinflammatory cytokines in macrophages[14], latent HIV-1 activation[51], and CD4+ T-cell apoptosis[52]. Here, we find that exosomes released from HIV-1-infected T cells promote proliferation, migration, invasion, as well as proto-oncogene expression of HNSCC and lung cancer cells in vitro and stimulate xenograft tumor growth in vivo. Most importantly, our findings indicate that HIV-infected patients under ART treatment contain circulating pro-tumor exosomes and that HIV-specific exosome cargo components contribute to the tumor-promoting effect. Exosomes from HIV-1-infected T cells contain HIV TAR RNA in vast excess over all viral mRNAs, caused by neither viral contamination nor RNA present outside of the exosomes[13]. We demonstrate that the exosomal TAR RNA, but not Tat or Nef RNA/protein, is specifically associated with cancer cell proliferation as well as expression of proto-oncogenes and ISGs. The correlation between abundance of TAR RNA loaded circulating exosomes and viral load/CD4+ T-cell counts can be critical in determining progression of certain NADCs, such as HNSCC and lung cancer. While our discoveries indicate that HIV-positive cancer patients contain higher levels of TAR RNA-bearing blood exosomes for cancer progression, the research does not support a casual role of exosomes loaded with TAR RNA in cell transformation and carcinogenesis. The higher incidence rate of HNSCC in HIV-infected patients is potentially attributed to increased infection of high-risk human papillomavirus and an increase in smoking in the population. Smoking is also a potent causal factor for lung cancer. Sampey et al.[14] and our results show that in patients with virtually undetectable virion levels, TAR RNA can still be found in blood exosomes. Therefore, our discovery indicates that TAR RNA-bearing exosomes in the circulation of HIV-positive patients act as a potential risk factor for the development and progression of some NADCs and presents a new arena of NADC research.

TAR RNA contains the stem–bulge–loop structure[13,53] and may act as a dsRNA to interact with cellular dsRNA binding proteins, including dsRNA-regulated protein kinase (PKR), the HIV TAR RNA-binding protein and La autoantigen[14,54,55]. It has been reported that the intact TAR RNA molecule is able to bind to PKR and TLR3 effectively, whereas the 5' and 3' stems (TAR microRNAs) bind best to TLR7 and TLR8 but not to PKR[14]. We show that the mutant TAR RNA with 5-nucleotide substitutions in the bulge and loop sequences, which retains the same stem structure as wild-type TAR RNA[35], cannot induce expression of *FOS* and *DEFB103* in cancer cells. The R06 RNA aptamer, which creates an imperfect hairpin and contains a 5'-GUCCCAGA-3' consensus motif complementary to the entire TAR loop to block the function of TAR RNA[36], blocks TAR RNA-induced gene

expression. In addition, transfection of the R06 aptamer into HIV-infected T-cell exosomes attenuates gene expression. Our findings indicate that the bulge–loop region of TAR RNA is critical for the pro-tumor function of HIV TAR RNA-bearing exosomes.

Cancer and stromal cell exosomes can modify the tumor microenvironment and immune response that favor cancer progression[56–59]. Boelens et al.[58] have reported that stromal cells release exosomes to target a subset of aggressive, triple-negative breast cancer to express ISGs, rendering their therapy resistance. Exosomes derived from latently EBV-infected lymphotropic cells contain small RNAs EBER1 and EBER2 which act as TLR3 ligands to trigger ISG expression in recipient cells[48]. Similarly, exosomes from HIV-infected T cells stimulate HNSCC cells to express ISGs. However, our results show that the exosomal HIV TAR RNA not only potentially triggers the proinflammatory TLR3 pathway, but also directly enhances proto-oncogene expression, proliferation, and migration of HNSCC cells through phosphorylation of ERK1/2 in an EGFR/TLR3-dependent manner. In addition, exosomes released from either EBV+ or EBV− cells fails to enhance HNSCC cell proliferation and proto-oncogene expression. Therefore, our findings indicate that the pro-tumor effect of HIV-infected T-cell exosomes on HNSCC and lung cancer cells is TAR RNA specific.

EGFR overexpression and its aberrant activity occur in over 90% of HNSCC cases and represent an independent prognostic marker correlating with increased tumor size, decreased radiation sensitivity, and increased risk of recurrence[39]. Given that more than 60% of non-small cell lung carcinomas express EGFR, the receptor has become an important therapeutic target for the treatment of these tumors[60]. EGFR activation by its ligands leads to effector kinase activation and endocytosis of the ligand/receptor complexes to early endosomes, where EGFR remains ligand bound, phosphorylated, and active until late stage of endosomal trafficking[61]. Activation of endosomal EGFR in association with TLR3 has been reported. Internalization of EGFR is involved in TNFα-induced EGFR endocytosis and intracellular signaling[62]. Activation of TLR3 by exogenous dsRNA leads to binding of TLR3 with EGFR and Src kinase residing in endosomes, resulting in TLR3 phosphorylation and recruitment of an adaptor protein TRIF (Toll-interleukin-1 receptor domain-containing adaptor protein inducing interferon-β) to induce *IFIT1* and *IFNB1* expression[26]. Our data indicate that TAR RNA-bearing exosomes activate the ERK cascade via interaction with EGFR and TLR3, a process that occurs in endosomes in association with TAR RNA.

Narayanan et al.[13] suggested that exosome TAR RNA could be introduced into recipient cells through fusion. However, our data show that EGFR mediates prompt entry of T-cell exosomes into recipient cells and that blockade of EGFR by a monoclonal antibody to the receptor significantly delays but not completely blocks exosome entry. EGFR-mediated entry of HIV-positive exosomes into target cancer cells is a necessary step for subsequent activation of ERK1/2, a process that involves EGFR and TLR3.

The EGFR/TLR3 axis activated by TAR RNA is specific for carcinoma cells that express EGFR, since B-cell lymphoma cells that lack EGFR do not respond to the HIV-positive exosomes for proliferation. Therefore, our discovery indicates that TAR RNA-bearing exosomes participate in promoting growth and progression of some, but not all, NADCs.

The tumor microenvironment contains surrounding blood vessels, immune cells, lymphocytes, fibroblasts, inflammatory cells, and the extracellular matrix[63]. Although the role of tumor-infiltrating lymphocytes (TILs) in cancer prognosis is not clear, the individual cancer specimens, including HNSCC and lung cancer, contain TILs at various densities[64,65]. Because HIV infects primarily T cells and myeloid cells in humans, HIV-infected TILs can release TAR RNA-bearing exosomes potentially to stimulate

cancer cell proliferation and establish the inflammatory tumor microenvironment, thus enhancing growth and progression of HNSCC and lung cancer. Our data indicate the presence of TAR RNA-containing cells/exosomes in the tumor niche of HIV-positive HNSCC patients. However, cancer cell exosomes may not augment HIV transactivation and they are not likely to have a direct impact on the HIV reservoir in NADC patients. Taken together, our findings reveal a novel mechanism by which TAR RNA-bearing exosomes enter into recipient cancer cells via EGFR, resulting in activation of the ERK1/2 cascade and expression of proto-oncogenes and ISGs through interaction with EGFR and TLR3, leading to enhanced growth and progression of NADCs, such as HNSCC and lung cancer.

## Methods

**Ethics statement.** All experiments involving animals were approved by the Case Western Reserve University (CWRU) Institutional Animal Care and Use Committee. For human subject studies, written informed consent was obtained from all study participants according to protocol approved by the Human Subjects Institutional Review Board (IRB) at University Hospitals Cleveland Medical Center.

**Cell culture, antibodies, and reagents.** J1.1 and 8E5/LAV cell lines were obtained from the NIH AIDS Reagent Program. 2D10 and C22G cells were obtained from Dr. Jonathan Karn (Case Western Reserve University, USA). These cells were maintained in RPMI-1640 medium (HyClone Lab., Inc., Logan, UT) supplemented with 10% exosome-depleted FBS, which was prepared by ultracentrifugation of FBS (ThermoFisher Scientific, Waltham, MA) at $100,000 \times g$ for 16 h at 4 °C[18], followed by collecting supernatants without disturbing the pellet. SCC9 (CRL-1629), H1299 (CRL-5803), Jurkat (TIB-152), CEM (CCL-119), LLC1 (CRL-1642), SU-DHL-16 (CRL-2964), Toledo (CRL-1631), Pfeiffer (CRL-2632), NC-37 (CCL214), and IB4 (HB-1064) cell lines were obtained from American Type Culture Collection (ATCC, Manassas, VA) and maintained following the vendor's instructions. The human tongue squamous cell carcinoma HSC3 cell line[22] was a gift from Dr. Yoshihiro Abiko (Health Sciences University of Hokkaido, Japan). The human oral buccal squamous cell carcinoma cell line TR146[66] was obtained from Dr. Aaron Weinberg (Case Western Reserve University, USA). UM-SCC-104[67] cell line was purchased from EMD Millipore (Burlington, MA). All cancer cell lines were maintained in DMEM (HyClone) and T as well as lymphoma cell lines were maintained in RPMI-1640 medium (HyClone) supplemented with 10% FBS and Primocin (InvivoGen, San Diego, CA) that is active against mycoplasmas, bacteria, and fungi in a humidified incubator with 5% $CO_2$ at 37 °C. The Tg26 HIV-transgenic mice (FVB background) were obtained from Dr. Leslie Bruggeman (Cleveland Clinic Lerner College of Medicine CWRU, USA)[37,38].

Antibodies against human CD9 (clone TS9, cat. no. 10626D, 1:500 dilution) and CD81 (clone M38, cat. no. MA1-10290, 1:1000) were purchased from ThermoFisher. Antibody against human CD63 (clone NKI/C3, cat. no. NBP2-32829, 1:200) was purchased from Novus Biologicals (Littleton, CO). Horseradish peroxidase (HRP)-conjugated goat anti-rabbit (cat. no. 1706515, 1:2000) and goat anti-mouse (cat. no. 1706515, 1:2000) IgG antibodies were obtained from Bio-Rad (Hercules, CA). Anti-pY1175-EGFR antibody was purchased from Millipore (ca. no. 05-483, 1:500). Anti-EGFR (cat. no. 4267, 1:1000), anti-phosphor-ERK1/2 (cat. no. 4376, 1:1000), anti-ERK1/2 (cat. no. 4696, 1:2000), anti-AKT (cat. no. 4691, 1:1000), anti-phosphor-AKT (cat. no. 4060, 1:2000), anti-pY1068-EGFR (cat. no. 2234, 1:1000), anti-pT669-EGFR (cat. no. 8808, 1:1000), anti-pS536-NFκB p65 (cat. no. 3033, 1:1000), anti-CD19 (cat. no. 90176, 1:1000), and anti-TLR3 (cat. no. 6961, 1:1000) antibodies were purchased from Cell Signaling Tech. (Danvers, MA). Recombinant TNFα and PMA were purchased from Sigma-Aldrich (St. Louis, MO). HIV TAR RNA, mut-TAR, R06 aptamer, and scrambled R06 aptamer were synthesized by the Integrated DNA Technologies (IDT Inc., Coralville, IA): HIV TAR RNA, 5'-GGUCUCUCUGGUUAGACCAGAUUUGAGCCUGGGGAGCUC UCUGGCUAACUAGGGAACC; mut-TAR, 5'-GGUCUCUCUGGUUAGACCA GAGGGAGCGAUUGGAGCUCUCUGGCUAACUAGGGAACC (mutated nucleotides are underlined); R06 aptamer, 5'-UCAACACGGUCCCAGACGUGU UGA; scrambled R06 aptamer, 5'-AGCAUUGGUACAAGCCAUCGCCGU. For RNA transfection, cells were cultured in 24-well plates, followed by RNA transfection using the X-tremeGENE HP Transfection Reagent (Roche Life Sci., Belmont, CA) following the manufacture's instruction.

**Patient plasma preparation.** Written informed consent was obtained from all study participants according to the protocol approved by the Human Subjects Institutional Review Board (IRB) at University Hospitals Cleveland Medical Center, Cleveland, OH. Whole blood was collected from controls and patients in heparinized tubes and then centrifuged down. Plasma was collected from tubes and placed in 2 ml cryotube aliquots. Then, 2 ml of the whole blood was put into 50 ml polystyrene tubes, diluted 1:2 with RPMI (BioWhittaker, cat. no. 12-167Q), underlayed with 10 ml of Ficoll-Paque Plus (GE Healthcare, cat. no. 17-1440-03).

Tubes were then centrifuged at 1600 rpm for 20 min at room temperature. The white blood cell layer on top of the Ficoll, containing lymphocytes and monocytes, was transferred into new 50 ml tubes. Remaining cell-free plasma was used for purification of exosomes.

**Exosome preparation and quantification.** Exosomes were prepared from cell supernatants by differential ultracentrifugation with filtration steps[18]. Briefly, cell culture media were centrifuged at $400 \times g$ for 5 min to remove cells. Then, the supernatants were spun at $11,000 \times g$ for 10 min to remove any possible apoptotic bodies and large cell debris. The supernatants were filtered through 1.2 μm nylon filters (ThermoFisher Sci.), followed by ultracentrifugation at $100,000 \times g$ for 90 min at 4 °C (50.2Ti rotor, Beckman Coulter, Brea, CA). The exosomes were suspended in 10 ml phosphate-buffered saline (PBS) and pelleted again by ultracentrifugation at $100,000 \times g$ for 90 min. Isolated exosomes were maintained at −80 °C in DMEM. To isolate exosomes from plasma, 2 ml of plasma was centrifuged at $400 \times g$ for 15 min to remove blood cell contaminants. The same volume of PBS (Lonza, Portsmouth, NH) was added to the supernatant, followed by centrifugation at $11,000 \times g$ for 10 min. After filtration through a 1.2 μm nylon filter, exosomes were pelleted by ultracentrifugation at $100,000 \times g$ for 90 min at 4 °C, washed with 10 ml of PBS, and pelleted again at $100,000 \times g$ for 90 min. Exosomes were quantified using the AChE assay system (System Biosci. Inc/SBI, Palo Alta, CA) following the manufacturer's instructions. Briefly, 20 μl resuspended exosomes were mixed with 80 μl of Exosome Lysis Buffer to extract exosome proteins. After centrifugation at $1500 \times g$ to remove debris, supernatants were mixed with the same volume of AChE Reaction Buffer on a microtiter plate, incubated at room temperature for 20 min, and then read on a microplate reader at 405 nm. Exosomes were quantified as numbers of exosomes per ml. Fluorescently labeled exosomes were prepared using the Exo-Red labeling kit (SBI) following the manufacture's instruction.

**Cell proliferation, migration, and invasion.** For cell proliferation analysis, cancer cells (5000) were plated in 96-well plates in triplicate and cultured in serum-free DMEM, pretreated with cetuximab (20 μg ml$^{-1}$) or U0126 (5 μM) for 30 min if necessary. Exosomes were then added into each well. Cells were labeled using Cell Proliferation Kit (MTT (3-[4,5-dimethylthiazol-2-yl]-2,5-diphenyltetrazolium bromide)) (Roche) and optical density (OD, 590 nm) values were measured every day for 4 to 5 days. Cell proliferation is presented as A590 nm. For detection of cancer cell migration, cancer cells were seeded in 24-well plates, cultured in serum-free DMEM, and scratches made using 200 μl plastic tips followed by wash with PBS. Cells were then treated with exosomes for 16 h. Cell migrations were recorded by microphotograph and the gaps were measured. Four microphotographs were taken from each well and the width of gaps was averaged. Each cell migration assay was conducted in triplicate. For cell invasion assays, cells (50,000) were plated in the upper chamber of 24-well BioCoat Matrigel Invasion Chambers (8 μm pore size, Corning, Bedford, MA) and then pretreated with cetuximab (20 μg ml$^{-1}$) or U0126 (5 μM) for 30 min if necessary. Exosomes isolated from supernatants of Jurkat, J1.1, 2D10, or C22G cells were added into the upper chamber of each well at indicated concentrations, respectively, for 18 h. Inserts were fixed with fixative (Differential Quick Staining Kit, Electron Microscopy Sciences Inc., Hatfield, PA) for 1 min, followed by Blue/Azure Dye (Richard-Allan Scientific, Kalamazoo, MI) staining for 10 min for visualization and cell counting. Four random fields were counted per well and the average number of migrated cells per field was calculated.

**Animal model.** All animal studies were approved by the Institutional Animal Care and Use Committee of the Case Western Reserve University in accordance with the guideline of National Institute of Health of the Unites States. Nude mice (*nu/nu*) were obtained from the Jackson Laboratory (Bar Harbor, ME) at 3–4 weeks old, with equal numbers of males and females. Exosomes were mixed with UM-SCC-104 ($5 \times 10^6$), HSC3 ($4 \times 10^6$), or H1299 ($5 \times 10^6$) cells in 200 μl of PBS at indicated concentrations. Each animal was injected with the cell/exosome mixture subcutaneously in two sites on opposite side of the dorsum of the anterior part of the body. Six animals were used for each arm of experiments. Animals were randomized for tumor cell injection. Tumor sizes were measured daily after injection. Xenograft tumors were harvested and weighted 13–14 days after inoculation.

**RT-PCR and real-time quantitative RT-PCR.** Total RNA was extracted from cells or exosomes using the High Pure RNA Isolation Kit (Roche Life Sci.) following the manufacture's protocol as described previously[68]. Cells grown in 6-well plates were lysed using the Lysis-/Binding buffer and the cellular lysates were centrifuged through the Filter Tube. The filter was then treated with DNase I, washed with Wash Buffer I/II, and the RNA was eluted using Elution Buffer. Total RNA samples were quantified using a spectrophotometer at $A_{260}$ and samples with the $A_{260}/A_{280}$ ratio ≥1.8 were used. For reverse transcription, total RNA (1 μg) was used for the first-strand complementary DNA (cDNA) synthesis using the Applied Biosystems High Capacity RNA-to-cDNA kit (ThermoFisher) in a total volume of 20 μl according to the manufacturer's instructions. For RT-PCR analysis, the cDNA (2 μl) was used in 25 μl of PCR amplification using *Taq* DNA polymerase (Invitrogen,

Carlsbad, CA) with appropriate primers. RT-PCR products were analyzed on 2% agarose gels with the GeneRuler 100 bp DNA Ladder (ThermoFisher). Quantitative PCR analysis was performed using Power SYBR Green PCR Master Mix (ABI) and detected by the Real-time PCR System (SteponePlus, ABI). Amplification was performed at 40 cycles of 94 °C for 15 s followed by 60 °C for 1 min. Quantification was determined using the comparative $\Delta\Delta C_T$ method and relative quantification was determined for each sample by normalizing with respective GAPDH expression[27]. Each qPCR was run in triplicate and each experiment was repeated at least 3 times. Primers for RT-PCR and quantitative RT-PCR are listed in Supplementary Table 4[26,27,48,67,69,70].

**Immunoblotting**. Total exosome proteins were purified using the Total Exosome RNA & Protein Isolation Kit (ThermoFisher) following the manufacturer's instructions. To prepare total cellular proteins, cells grown in 6-well plates were washed with PBS and then cellular lysates were obtained by adding 150 μl of RIPA Lysis and Extraction Buffer (ThermoFisher). Protein lysates were separated by sodium dodecyl sulfate–polyacrylamide gel electrophoresis and then transferred onto polyvinylidene fluoride membranes (Millipore) for immunoblot analysis. The membranes were blocked in 5% non-fat milk for 1 h at room temperature, followed by incubation with primary antibodies overnight at 4 °C. Then, the membranes were washed and incubated with appropriate HRP-conjugated secondary antibodies for 1 h at room temperature. Protein detection was performed by chemiluminescence using an ECL kit (ThermoFisher) with the ChemiDoc XRS+ Imaging System (Bio-Rad). Uncropped scans of important blots are presented in the Supplementary Information.

**TLR3-knockout cells**. TLR3 Double Nickase Plasmid (h) (cat. no. SC-400512-NIC, Santa Cruz Biotech., Dallas, TX) was used to knockout TLR3 in HSC3 cells with UltraCruz Transfection Reagent (cat. no. SC-395739, Santa Cruz) following the manufacturer's instructions. Briefly, HSC3 cells were seeded into 6-well plate and reached 80% confluence. Then, 1 μg plasmid and 5 μl of transfection reagent were used for each transfection. After 2 days of transfection, cells were selected with 1 μg ml$^{-1}$ puromycin (Sigma-Aldrich) in 10 cm dish for 2 days, and grown in regular culture medium. Well-isolated clones were picked and tested for TLR3 level with immunoblotting.

**Imaging analysis of exosome entry**. HEK293 cells (CRL-1573, ATCC) were transiently transfected with EGFR-GFP (Addgene, Cambridge, MA). Exosomes isolated from J1.1 cells were labeled with the Exo-Red Labeling kit (SBI) following the manufacturer's instructions. Transfected HEK293 cells were seeded into glass bottom 35 mm dishes. Imaging was performed with the Deltavision RT epifluorescent microscope system (GE Healthcare Life Sciences, Marlborough, MA). The scope was fitted with an automated stage (GE Healthcare Life Sciences) and a temperature-controlled chamber. The time-lapse POL (TRANS, to show structure of cells) and red fluorescent (to show exosomes) images in z-series were captured simultaneously. Out-of-focus light was digitally removed using the software deconvolution function. The 3D volume projections were generated using the same software and avi format movies were exported and converted to mov format movies using the Movie Maker software (Microsoft Inc., Redmond, WA).

**Statistical analysis**. Results of treatments were compared with those of respective controls. Error bars represent ± s.d. Cell proliferation data were subjected to one-way analysis of variance (ANOVA) when sample sizes were $n \leq 3$. Statistical significance was considered at $p < 0.05$. For data with $n \leq 5$, F-test was applied. For $n \geq 6$, two-tailed paired Student's t-test with two-sample equal variance for comparison of two groups was applied. The $p \leq 0.05$ was considered to be statistically significant. Data analyses were performed and graphs were generated using Prism (GraphPad Software, La Jolla, CA) and Excel 2013 (Microsoft).

## Data availability

All data that support the findings of this study are available from the corresponding author upon request.

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

## Acknowledgements

We thank Drs. Ganes Sen and Pamela Wearsch for helpful discussion, Dr. Claudia Cabrera for assistance in statistics, the Cytometry & Imaging Microscope Shared Resource of the Case Comprehensive Cancer Center (P30CA043703) and the CWRU Molecular and Cellular Virology Core. HIV-1 LAV-infected Jurkat E6 cells (J1.1) and 8E5 cells were obtained through the National Institute of Health (NIH) AIDS Reagent Program (Dr. Thomas Folks), Division of AIDS, NIAID, NIH. HIV-positive head and neck cancer specimens were provided by AIDS and Cancer Specimen Resource (ACSR), funded by the National Cancer Institute (UM1CA181255). This work was supported by grants from the NIH/National Institute of Dental and Craniofacial Research (NIDCR) R01DE025284 (to G.J.), R01DE026925 (to G.J.), and a pilot award (to G.J.) from Case Comprehensive Cancer Center (CCCC) and the NIH-funded CCCC Support Grant P30CA043703. B.W. is supported by the NIH grants R01NS096956 and R01CA155676.

## Author contributions

L.C. and G.J. conceived and designed the experiments; L.C. designed and performed the most experiments and data analyses; Z.F. performed immunoblotting and exosome entry assays; H.Y. performed immunoblotting assays; D.B. and S.F.S. tested HIV and provided plasma samples of HIV-positive and HIV-negative individuals; U.M. and J.K. grew and provided 2D10 and C22G cells; L.B. and B.W. prepared and provided Tg26 mouse blood samples; C.Z. provided reagents; C.V.H., J.K., S.F.S., and B.W. provided advice on project design and edited the manuscript; G.J. directed and supervised the study and interpretation of data and prepared the manuscript.

## Additional information

**Competing interests:** The authors declare no competing interests.

