## [Peer Review File · Nature Communications]

Reviewers' Comments:

Reviewer #1:

Remarks to the Author:

This is an intriguing manuscript that presents a potentially important mechanism by which HIV-infected cells can promote tumorigenesis through HIV-TAR found in exosomes. The results can help explain the increased incidence of certain tumors (especially non-AIDS defining tumors) in patients with HIV infection. I did, however, have some general and specific comments and concerns:

1. The experiments with exosomes (such as Figure 2a), use a high concentration (4×10^9) exosomes. The experiments performed with exosomes from HIV-infected patients (like Figure 2e) used a similarly high concentration of exosomes. The question must be raised if these effects are only seen with an extremely high concentration of exosomes, or whether they could occur in patients. The authors should clearly address this and address the uncertainty regarding whether this effect can apply in patients. . Also, a dose-response curve should ideally be shown which would provide a sense of the concentration-dependence of the effect.
2. Much of the increase in head and neck squamous cell carcinoma (HNSCC) in HIV patients is from HPV-associated tumors, and much can be accounted for by an increase in smoking in this population. Also, epidemiologic evidence for a direct role of HIV in lung cancer is somewhat controversial. The findings here would be bolstered by looking at another tumor in which it is clearer that HIV plays a direct role in pathogenesis, such as B cell non-Hodgkin lymphoma.
3. More should be said about the HIV-infected patients from whom the exosomes were purified. Were they on cART? What was their viral load? Can the authors study exosomes from patients with both high and low viral loads?
4. In Figure 3j, the authors conclude that scrambled R06 aptamer does not suppress DEFB103 mRNA induction. However, there is a trend down ($p=0.08$), and one can question whether this represents a non-effect, especially when using a Student's t test with 3 replicates (not a truly proper use of Student's t test). This finding should be studied further, or the results re-interpreted with regard to the above comment.
5. Minor comment: In Supplement figure 4, lane 1 should be labeled. Also, more detailed explanation of this essentially negative figure should be provided.

Reviewer #2:

Remarks to the Author:

In this study, the authors claim that exosomes from both latently HIV-1-infected T cells or those from sera of HIV-positive subjects significantly promote proliferation and progression of HNSCC and lung cancer cells and that this effect is mediated by activation of the ERK cascade via HIV TAR RNA in the exosomes that upon transfer promotes tumor growth.

While in principle this is an interesting hypothesis, the in vivo data shown is not sufficiently conclusive to support the major claim. My largest concern, reducing enthusiasm for publication at this stage in Nat Comm., is the physiological relevance of the findings. The in vivo data (Fig. 2d) are the most important advance in their study but unfortunately not sufficiently developed or controlled. Moreover, prior reports, that were not cited (Boelens et al., Cell 2014; Baglio et al., PNAS 2016; Nabet et al., 2017), have shown that defined small RNA species in exosomes from virus and tumor cells activate PRRs in recipient cells, making the mechanistic advance of this study somewhat limited.

To evaluate the effect of HIV-1-infected T-cell exosomes on tumorigenesis in vivo, the authors inoculated nude mice subcutaneously with HNSCC cells mixed with the same concentration of exosomes from HIV-positive (HIV+) or HIV-negative (HIV-) control Jurkat cells, respectively. Tumors inoculated together with the J1.1

cell exosomes grew faster and exhibited significantly higher tumor weight compared to those mixed with Jurkat exosomes. There are many explanations possible for these results and technical issues require that these data are interpreted with caution,

- The authors do not show that the effects seen in vivo are dependent on TAR and/or TLR3/ERK cascade.
- Mixing tumor cells with exosomes upon inoculation do not recapitulate in vivo 'priming' of the tumor cells.
- Using Immune compromised mice, neglects the role of immune (stroma) cells.
- Ultracentrifugation of exosomes, leads to many artifacts through co-isolation of unwanted particles.
- To substantiate the claim that TLR3 is involved, can the authors inhibit this in vivo? Does this reduce tumor growth.
- It is unclear whether the amount of ctrl exosomes is the same, quantitation of exosomes remains an issue.
- Can the authors show that TAR is present in tumors of HIV+ patients?

Apart from the in vivo data, i feel the in vitro data is more convincing and better developed but as said, I'm not sure what the real mechanistic advance is here compared to prior publications on this topic. The finding that serum exosomes from IV patients have similar effects as purified in vitro TAR+ exosomes is very interesting but what i miss here is quantitative data. How much Tar is in the exosomes, how much is transferred, what is the entry mechanism of the exosomes in the tumor cells, is this tumor cell specific, do HIV proteins play a role (what i could imagine)? Can the authors reveal something about the stoichiometry? Such data would set it more apart from prior studies.

Reviewer #3:

Remarks to the Author:

The study suggests that exosomes from HIV-infected cells contain TAR RNA fragments to drive growth and cancer progression. This story builds in part on a wealth of previous studies e.g. on HIV-1 Tat as Kaposi's sarcoma inducer. All studies in artificial experimental systems should be carefully checked as the major reason of HIV-related cancers is simply the virus-induced immune suppression! I therefore did not like the first 2 sentences of the abstract. It is not really known at all if "residual and persistent HIV replication" is needed for such cancer induction. Antiviral drugs stop this cancer route, but they also resolve the immune suppression.

The study is presented in a confusing manner, e.g. the CEM control cell is presented on page 4, but only shows up in Fig 1B. Panel 1A lacks any of these controls. Panel 1B lacks an internal control. Later AChE is launched as control, but this should be explained before panel 1B. And why do we need to see two AChE titration in panels 1C and 1D. Only 1C is mentioned in the text (which probably should be 1D?). Anyhow, all very confusing.

One sometimes uses J1.1 cells with or without TNF treatment. Isn't a latent HIV infection mean that the integrated provirus is transcriptionally silent? One should then not expect any TAR transcript to be present.

Anyhow, it may be important to analyze some of the other J-LAT clones to beef up these findings. Do they all do the same thing.

Detailed proteomic and lipidomic analyses were done, but the results are not worked out or discussed.

The study should provide more information on items like exosome-free FBS and the quality of the differential centrifugation method.

Page 5: viral taxonomy?

The studies with patient samples is also not convincing at all. Very small patient numbers are used (3 versus 2) and the 2 control exosome preparations were mixed, but that was not done for the 3 patients. This is not correct as mixing will change the actual composition for differentially expressed items!

I understand the switch to synthetic TAR studies, although it is quantitatively difficult to compare synthetic RNA transfection with exosome delivery and one should always realize that the former method will likely test unnatural TAR amounts. Aptamers are used to demonstrate that TAR is also the critical component in exosomes, but the results did not convince me. The aptamer effect in Fig 3i is quite small and about the same as that of the mutated control aptamer.

Authors' Response

NCOMM-17-01488B

Response to Editor's comments:

We would expect any revised manuscript to provide further evidence that TAR in exosomes from patients under cART affects cancer development and to provide additional data to support its pathophysiological relevance as pointed by all three reviewers.

Authors' Response: We thank the Editor for pointing out the major concerns from all three reviewers. We have provided further experimental data and analyses to address this concern. We collected plasma samples from six HIV-positive patients under cART, six normal control individuals and five HIV-positive head and neck squamous cell carcinoma (HNSCC) patients treated with cART. We tested those plasma exosomes and found that all HIV-positive exosomes significantly promoted proliferation of cancer cells compared with those from normal controls and that all HIV-positive exosome samples contained TAR RNA. Taking together, our data indicate the pathophysiological relevance of TAR RNA-bearing exosomes with promotion of cancer growth and progression. We have made substantial revision on the issue by adding a new section (page 9) and a new figure (Figure 4) with new data to address this concern. Patient information, including genders, ages, cART treatment, viral load, and CD4 T cell counts was listed in Supplementary Table 1 and 2.

Response to reviewers' comments:

We thank reviewers for their constructive criticisms and comments. We have added further experimental data and analysis and substantially revised the manuscript to address each of reviewers' comments. Following is our response to the comments.

Reviewer #1, Expertise: HIV related cancers/viral related cancers (Remarks to the Author):

This is an intriguing manuscript that presents a potentially important mechanism by which HIV-infected cells can promote tumorigenesis through HIV-TAR found in exosomes. The results can help explain the increased incidence of certain tumors (especially non-AIDS defining tumors) in patients with HIV infection. I did, however, have some general and specific comments and concerns:

Authors' Response: We greatly appreciate the recognition of the potential importance of our discoveries by reviewer 1 and find the comments both insightful and helpful. Below is our point-by-point response to the reviewer's critiques.

1. The experiments with exosomes (such as Figure 2a), use a high concentration (4x10⁹) exosomes. The experiments performed with exosomes from HIV-infected patients (like Figure 2e) used a similarly high concentration of exosomes. The question must be raised if these effects are only seen with an extremely high concentration of exosomes, or whether they could occur in patients. The authors should clearly address this and address the uncertainty regarding whether this effect can apply in patients. Also, a dose-response curve should ideally be shown which would provide a sense of the concentration-dependence of the effect.

Authors Response: 1) We thought about this good question as well. Concentrations of exosomes used in our experiments were quantified using the EXOCET exosome quantification assay (System Biol Inc.), which converted acetylcholinesterase (AChE) activity to numbers of exosomes using the manufacturer's calibration standards. We quantified total exosomal proteins at various concentrations of exosomes and found that concentrations of exosomes used in our report was in the range of 60-180 µg/ml total exosomal proteins and reported the data in Supplementary Fig. 1c. Others have reported using exosomes with total exosomal proteins from 50-250 µg/ml (Lee J, et al. PLoS One, V.8(12); 2013; Higginbotham JN, et al. Curr Biol, 21(9); 2011; Zhang B, et al. Stem Cell Int. 2016: 1929536). Therefore, the concentrations that we used are in the regular range for studies involving exosomes. The amount of exosomes at 4×10^9 /ml in Fig 2 was equivalent to ~80 µg/ml of total exosomal protein. Therefore, the pro-tumor effect of HIV-infected T-cell exosomes was not due to extremely high concentrations of exosomes.

2) The concentration of exosomes in patients can vary both spatially and temporarily. In this revised report, we typically obtained $30\sim 40 \times 10^9$ exosomes from 2ml of patient plasma, which was about $15\sim 20 \times 10^9$ exosomes/ml plasma, indicating that we were able to readily isolate significant amount of exosomes from HIV patient plasma. In the new results that we obtained since the initial submission, we were also able to detect TAR RNA in the FFPE tissues of HIV-positive oral squamous cell carcinoma patients, the sources of which can be either circulation, or local infiltrating T cells, or both. Therefore, it is very likely that tumor cells can encounter the level of exosomes that we used in this study.

2. Much of the increase in head and neck squamous cell carcinoma (HNSCC) in HIV patients is from HPV-associated tumors, and much can be accounted for by an increase in smoking in this population. Also, epidemiologic evidence for a direct role of HIV in lung cancer is somewhat controversial. The findings here would be bolstered by looking at another tumor in which it is clearer that HIV plays a direct role in pathogenesis, such as B cell non-Hodgkin lymphoma.

Authors Response: 1) The reviewer's point is valid. Indeed, infection with high-risk HPV and smoking are major causal factors for HNSCC in both HIV-positive and HIV-negative populations. However, a most recent epidemiological study reported by Mahale et al. indicated that standardized incidence ratios (SIRs) significantly increased for HNSCC (SIR=1.66) and lung cancer (SIR=1.71) in HIV-infected subjects. Excess absolute risks (EARs) increased with age for lung cancer and HNSCC, among other cancers (Mahale et al., Clin Infect Dis, 2018. PMID: 29325033). In addition, Shiels et al. provided epidemiological evidence showing that lung cancer, anal cancer, HNSCC and myeloma were diagnosed at modestly younger ages, possibly reflecting accelerated cancer progression (Shiels, et al., Clin Infect Dis, 64:468-, 2017. PMID: 27940936). Therefore, our data reported in this manuscript were supported by these epidemiological evidence.

2) As suggested by the reviewer, we treated three B cell non-Hodgkin's lymphoma (an AIDS-defining cancer) cells (CD19+) with exosomes from HIV-infected T cells for cell proliferation and found that HIV-positive exosomes were not able to further enhance cell growth compared with control T-cell exosomes. Further investigation found that these B cell lymphoma cells lack EGFR, which was critical for HIV-positive exosome induced HNSCC and lung cancer progression. These new results were presented in Fig. 5d and 5e of the revised manuscript to support the concept that HIV-positive exosomes played a critical pathogenesis role in cancers with EGFR as a driver oncogene.

3. *More should be said about the HIV-infected patients from whom the exosomes were purified. Were they on cART? What was their viral load? Can the authors study exosomes from patients with both high and low viral loads?*

Authors' Response: Supplementary Table 1 and 2 listed HIV-infected patient information, including genders, ages, viral load, treatment, and CD4 counts when patients were recruited. Our results included exosomes from patients with both high and low viral loads; exosomes purified from all HIV-positive patients were able to induce HNSCC and lung cancer cell proliferation. We have made substantial revision on the issue by adding a new section and a new figure (Fig. 4) with new data to address this concern. Furthermore, our results showed that exosomes purified from latent or TNFalpha HIV-positive J1.1 cells promoted cancer cell proliferation and migration at the same level, suggesting that viral loads were not the determining factor for exosomes to promote tumor growth and progression.

4. *In Figure 3j, the authors conclude that scrambled R06 aptamer does not suppress DEFB103 mRNA induction. However, there is a trend down ($p=0.08$), and one can question whether this represents a non-effect, especially when using a Student's *t* test with 3 replicates (not a truly proper use of Student's *t* test). This finding should be studied further, or the results re-interpreted with regard to the above comment.*

Authors Response: We have repeated the experiments at least three times with modification in transfection and found that transfection of R06 to exosomes from HIV-infected T cells significantly reduced ability of HIV+ exosomes to enhance expression of the DEFB103 gene. We agreed with the reviewer that the Student's *t* test was not proper for statistical analysis in this case. In this revision we presented data (means \pm SD, $n = 3$; one experiment out of three repeats) with *F*-test, which fits for replicates fewer than 5.

5. *Minor comment: In Supplement figure 4, lane 1 should be labeled. Also, more detailed explanation of this essentially negative figure should be provided.*

Authors Response: Lane 1 of the figure has been labeled. Supplementary Fig. 4 has been moved to new Fig. 4. More detailed explanation has been provided in Result Section (page 9 and 10).

Reviewer #2, Expertise: exosomes (Remarks to the Author):

In this study, the authors claim that exosomes from both latently HIV-1-infected T cells or those from sera of HIV-positive subjects significantly promote proliferation and progression of HNSCC and lung cancer cells and that this effect is mediated by activation of the ERK cascade via HIV TAR RNA in the exosomes that upon transfer promotes tumor growth.

While in principle this is an interesting hypothesis, the in vivo data shown is not sufficiently conclusive to support the major claim. My largest concern, reducing enthusiasm for publication at this stage in Nat Comm., is the physiological relevance of the findings. The in vivo data (Fig. 2d) are the most important advance in their study but unfortunately not sufficiently developed or controlled. Moreover, prior reports, that were not cited (Boelens et al., Cell 2014; Baglio et al., PNAS 2016; Nabet et al., 2017), have shown that defined small RNA species in exosomes from virus and tumor cells activate PRRs in recipient cells, making the mechanistic advance of this study somewhat limited.

Authors' Response: We greatly appreciate the reviewer's valuable comments. To fully address the reviewer's largest concern, multi-pronged efforts were undertaken to demonstrate the physiological relevance of TAR RNA in human patients.

1) We have doubled HIV+ patient plasma exosome samples and detected TAR RNA in all of them.

2) In addition, since the last submission, we have obtained plasma from HIV+ HNSCC patients and detected TAR RNA in all of them as well.

3) Moreover, we have acquired FFPE sections from HIV+ HNSCC biopsies. Remarkably, we were able to detect TAR RNA from RNA extracted from these biopsy tumor sections, which not only provided support for disease relevance of our study, but also pointed to the potential of using this technology for future identification of patients who may benefit from novel therapies targeting TAR RNA.

4) In this revised manuscript, we discussed studies reported by Boelens et al., Baglio et al. and Nabet et al. on page 18, line 311 to 319. Our results show a novel mechanism by which the exosomal HIV TAR RNA not only represents exogenous pathogen-associated molecular pattern (PAMP) signals to potentially trigger the pro-inflammatory TLR3 pathway, but also directly enhance proliferation and progression of cancer cells through ERK1/2 phosphorylation in an EGFR-dependent manner. Interestingly, EGFR mediates rapid entry of exosomes into recipient cells, together with TLR3 and the TAR RNA, to stimulate activation of the ERK1/2 cascade and expression of ISGs.

To evaluate the effect of HIV-1-infected T-cell exosomes on tumorigenesis in vivo, the authors inoculated nude mice subcutaneously with HNSCC cells mixed with the same concentration of exosomes from HIV-positive (HIV+) or HIV-negative (HIV-) control Jurkat cells, respectively. Tumors inoculated together with the J1.1 cell exosomes grew faster and exhibited significantly higher tumor weight compared to those mixed with Jurkat exosomes. There are many explanations possible for these results and technical issues require that these data are interpreted with caution.

- The authors do not show that the effects seen in vivo are dependent on TAR and/or TLR3/ERK cascade.

Authors' Response: We generated TLR3-knockout HNSCC cells and inoculated together with exosomes from HIV-infected or control T cells into nude mice. While HIV+ T cell exosomes significantly promoted TLR3 wild type HNSCC xenograft tumor growth, the exosomes failed to induce growth of xenograft tumors inoculated with TLR3-KO cancer cells, indicating that HIV-infected T-cell exosomes enhanced tumor growth in vivo was TLR3-dependent (now in Fig. 6e).

- Mixing tumor cells with exosomes upon inoculation do not recapitulate in vivo 'priming' of the tumor cells.

Authors' Response: The reviewer raised a valid point. However, in the absence of better model this approach was as close as a simulation in vivo, as to do systemic treatment will require far more much HIV+ exosomes to be feasible. In addition, our results suggested that HIV-infected T-cell exosomes were risk factors for growth and progression of existing tumors.

- Using Immune compromised mice, neglects the role of immune (stroma) cells.

Authors' Response: Although it will be better to use immune competent mice as the reviewer suggested, with human cancer cell xenograft model, we had to use immune deficient mice.

- Ultracentrifugation of exosomes, leads to many artifacts through co-isolation of unwanted particles.

Authors Response: We have perfect control of NON-HIV exosomes prepared by the same method. In addition, cells were maintained in media supplemented with exosome-depleted FBS. The differential ultracentrifugation method removed some unwanted particles, such as apoptotic bodies, at high speed.

- To substantiate the claim that TLR3 is involved, can the authors inhibit this in vivo? Does this reduce tumor growth?

Authors Response: Yes, we generated TLR3-null HNSCC cells and inoculated together with exosomes from HIV-infected or control T cells into nude mice. We found that HIV+ T cell exosomes did not promote TLR3-null tumor growth (in revised Fig. 6e).

- It is unclear whether the amount of ctrl exosomes is the same, quantitation of exosomes remains an issue.

Authors' Response: We did control the amount of exosomes to be the same between HIV+ and HIV- preps for all experiments using the method shown in Fig. 1.

- Can the authors show that TAR is present in tumors of HIV+ patients?

Authors' Response: Thanks for the suggestion by the reviewer. Yes, we have detected TAR in 1) formalin-fixed paraffin-embedded (FFPE) biopsy specimens of HIV-positive HNSCC patients, 2) plasma exosomes of HIV-positive HNSCC patients. These results have been present in new Fig. 4d, e, f, relative patient information in supplementary table 1 and 2.

Apart from the in vivo data, i feel the in vitro data is more convincing and better developed but as said, I'm not sure what the real mechanistic advance is here compared to prior publications on this topic. The finding that serum exosomes from HIV patients have similar effects as purified in vitro TAR+ exosomes is very interesting but what i miss here is quantitative data.

How much Tar is in the exosomes, how much is transferred.

Authors' Response: We quantified TAR RNA in exosomes purified from plasma of HIV-infected subjects and compared it with exosomal TAR RNA from 8E5 cells, which contained a single copy of HIV genome. Our results showed that HIV-patient plasma exosomes contained 7~80-fold higher TAR RNA than that in 8E5 exosomes (in revised Fig. 4d, e).

What is the entry mechanism of the exosomes in the tumor cells?

Authors' Response: We found that entry of exosomes into tumor cells was EGFR dependent, because treatment of cancer cells with the humanized monoclonal antibody to EGFR blocked entry of fluorescently labeled exosomes into recipient cells. These data are now presented as Supplementary Movie 1 and Supplementary Fig. 5 in the revised manuscript.

Is this tumor cell specific, do HIV proteins play a role (what i could imagine)? Can the authors reveal something about the stoichiometry? Such data would set it more apart from prior studies.

Authors' Response: Yes. Specifically, it is carcinoma cell specific and requires EGFR of cancer cells. HIV proteins did not play a role because exosomes from HIV+ C22G T cells, which contain a defective HIV Tat and HIV Nef-null, could still induce cancer cell proliferation and migration. To address the concern about stoichiometry, we did proliferation assays of cancer cells in response to different concentrations of HIV-infected T-cell exosomes. The data indicated that cancer cells responded to HIV-infected T-cell exosomes in a dose-dependent fashion (Supplementary Fig. 2c).

Reviewer #3, Expertise : HIV/TAR infection (Remarks to the Author):

The study suggests that exosomes from HIV-infected cells contain TAR RNA fragments to drive growth and cancer progression. This story builds in part on a wealth of previous studies e.g. on HIV-1 Tat as Kaposi's sarcoma inducer. All studies in artificial experimental systems should be carefully checked as the major reason of HIV-related cancers is simply the virus-induced immune suppression! I therefore did not like the first 2 sentences of the abstract. It is not really known at all if "residual and persistent HIV replication" is needed for such cancer induction. Antiviral drugs stop this cancer route, but they also resolve the immune suppression.

Authors' Response: The reviewer was correct in pointing out that HIV-related cancers is simply the virus-induced immune suppression, which is a causal factor for AIDS-defining cancers, including Kaposi's sarcoma, cervical cancer and Non-Hodgkin lymphoma.

However, the central goal of this manuscript is to elucidate the mechanisms of none-AIDS-defining cancers among HIV-infected people under cART, who do become susceptible to the development of non-AIDS-defining cancers, including HNSCC and lung cancer.

We agree with the reviewer's comments on "residual and persistent HIV replication". The statement has been changed to "human immunodeficiency virus pathogenesis plays key roles..." Even cART dramatically decreased AIDS-defining cancers and resolved the immune suppression, exosomes derived HIV-infected cells can still promote tumor growth and progression in non-AIDS-defining cancer patients.

The study is presented in a confusing manner, e.g. the CEM control cell is presented on page 4, but only shows up in Fig 1B. Panel 1A lacks any of these controls. Panel 1B lacks an internal control. Later AChE is launched as control, but this should be explained before panel 1B. And why do we need to see two AChE titration in panels 1C and 1D. Only 1C is mentioned in the text (which probably should be 1D?). Anyhow, all very confusing.

Authors' Response: We have corrected and/or clarified these points in the revised manuscript (new Fig. 1) on page 5.

One sometimes uses J1.1 cells with or without TNF treatment. Isn't a latent HIV infection mean that the integrated provirus is transcriptionally silent? One should then not expect any TAR transcript to be present.

Authors' Response: Extensive previous studies by Kashanchi's and other groups as well as our own data show that latently HIV-infected T cells can express TAR in cells and in exosomes, in excess of any other HIV RNAs.

Anyhow, it may be important to analyze some of the other J-LAT clones to beef up these findings. Do they all do the same thing?

Authors' Response: We used three HIV-infected Jurkat clones and an HIV-infected CEM cell line. They all express TAR in exosomes and promote cancer cell proliferation and progression.

Detailed proteomic and lipidomic analyses were done, but the results are not worked out or discussed.

Authors' Response: The full characterization of the proteins and lipids is beyond the scope of this study. The data have been removed.

The study should provide more information on items like exosome-free FBS and the quality of the differential centrifugation method.

Authors' Response: We thank the reviewer for pointing this out. We have provided more information about methodology of exosome-free FBS and quality of the differential ultracentrifugation method in revised manuscript (page 5 and 19).

Page 5: viral taxonomy?

Authors' Response: Since proteomics and lipidomics data have been removed from the revised manuscript, viral taxonomy is no longer needed.

The studies with patient samples is also not convincing at all. Very small patient numbers are used (3 versus 2) and the 2 control exosome preparations were mixed, but that was not done for the 3 patients. This is not correct as mixing will change the actual composition for differentially expressed items!

Authors' Response: We agree with the reviewer. We have provided further experimental data and analyses to address this concern. We collected plasma samples from six HIV-positive patients under cART, six normal control individuals and five HIV-positive HNSCC patients treated with cART. We tested each of those plasma exosomes and found that all HIV-positive exosomes significantly promoted proliferation of cancer cells compared with those from normal controls and all HIV-positive exosome samples contained TAR RNA. We have made substantial revision on the issue by adding a new section (page 9) and a new figure (Fig. 4) with new data to address this concern. Patient information, including genders, ages, cART treatment, viral load, and CD4 T cell counts was listed in Supplementary Table 1 and 2.

I understand the switch to synthetic TAR studies, although it is quantitatively difficult to compare synthetic RNA transfection with exosome delivery and one should always realize that the former method will likely test unnatural TAR amounts. Aptamers are used to demonstrate that TAR is also the critical component in exosomes, but the results did not convince me. The aptamer effect in Fig 3i is quite small and about the same as that of the mutated control aptamer.

Authors' Response: TAR RNA transfection provided supporting data for its major role. We believe that we have strong evidence showing the role of TAR RNA in promoting tumor growth and progression. We now further provided data to show that cancer cells responded to HIV-infected T-cell exosomes in a dose-dependent fashion (Supplementary Fig. 2c). Fig. 3i did show significant effects of R06 aptamer.

Reviewers' Comments:

Reviewer #1:

Remarks to the Author:

This manuscript has been improved since the original submission, and a number of the initial reviewers' comments addressed. The experiments on tumor samples and lymphomas are helpful in clarifying this phenomenon. As I indicated before, the potential role of exosomes from HIV-infected cells playing a role in tumor development in HIV patients is intriguing. However, some lingering concerns remain.

1. The question remains as to whether the concentration of the exosomes in the patients' plasma is really high enough to yield the tumor effects reported. The new data on tumor biopsies is somewhat helpful, but it is again unclear just how much TAR was present (this was RT-PCR which can be very sensitive). It is hard for me to tease this out -- in part, this is because the Figure legends often do not say much about the concentrations of exosomes used, and this should be spelled out so the reader can readily appreciate it. Also, in the response to the comment #1, the authors note that they typically obtained 15-20x10⁹ exosomes per ml of patient plasma and that the results in Figure 2 (I assume 2a) were obtained with 4x10⁹ exosomes per ml. If this is the case, then some of the key observations noted should be obtainable with unmodified patient plasma, and the effects potentially inhibited if exosome-depleted plasma is used. If this is the case, the paper would be strengthened by showing it. If not, or these calculations are incorrect, the authors should make it much clearer to the reader just how the effects seen relate to the concentrations in the patients' plasma and report the concentrations used throughout. In a similar way, it should be made clearer just how the animal experiments in Figure 2d were done.

a. In this regard, the authors say that the concentrations used is that used by previous authors (line 105), but it is unclear that the previous authors proved that these concentrations are biologically realistic. This is not, in itself, a sufficient argument.

2. A number of the experiments (see Fig 3) describe effects on DEFB103, and this is used as a surrogate for the tumor promoting effect. However, it is not clearly explained why this is a good surrogate. This should be spelled out, or else other endpoints directly related to tumor development used.

3. In the initial review, concern was raised that in Figure 3j, the scrambled R06 aptamer had some effect on suppressing DEFB103 (p<0.08). The authors note they have changed the statistical test used. I cannot go back and see the original figure, but it appears that this control has now just been removed. If so, this is concerning and the control should be shown.

4. I have a number of relatively minor points in how the article is written that should be addressed:

a. In first 2 sentences of abstract, HIV is believed to play a key role in CERTAIN (but not all NADC). Also, not all mechanisms are obscure -- some are well known.

b. It is not really appropriate to generalize the results here to all NADC. Only 2 cancers in which this effect is seen are shown. I would either say "certain" NADC, or else focus all the description in the paper on the cancers studied.

c. Line 46. The incidence of cervical cancer was not dramatically reduced by ART.

d. On line 79 and throughout the manuscript, the authors use the term "progression". This term is imprecise. The authors should instead use descriptions of the actual effects being studied (migration, etc.)

e. On line 293, the authors speculate that exosomes could be used as a "pathogenic biomarker" for cancer progression, but to do so, they would have to show that the concentration of such exosomes differed greatly among patients and was predictive. I would tone this down.

Reviewer #2:

Remarks to the Author:

The authors have addressed many concerns a few are remaining.

The authors show 'relative TAR levels', while the fold increases may seem physiologically relevant at a first glance fig 4d-f, I wonder what the levels would be compared to infected HIV cells? i.e. could the authors correlate their TAR levels in exosomes and the tumor tissues to 'infected cell equivalents'? The reason I ask is whether the authors can exclude that a single HIV-infected T cell infiltrating the tumor mass could be responsible for the signals observed.

I still wonder if the effects seen are HIV-TAR exosome 'specific'. Another natural TLR3 ligand is the human tumor virus related EBER1 molecule, also found in exosomes (Iwakiri et al., 2009 JEM, Baglio et al PNAS 2016). Would exosomes with EBERS have similar effects? Could the authors transfect wild-type Jurkat exosomes with EBER1 and show similar (in vitro results) on cancer cell proliferation or perhaps use LCL exosomes directly, assuming they also enter recipient cells via EGFR. This to me would provide more concrete evidence that exosomal small RNAs (from virus-infected cells) can have pro-tumorigenic effects mediated via TLR3.

Finally the in vivo data with the TLR3KO cells are compelling, however did the authors rule out a potential growth disadvantage of the KO cells?

Reviewer #3:

Remarks to the Author:

I still have a very major problem with the first half of the first sentence of the abstract. How do we know that HIV pathogenesis is contributing to NADCs? As correctly stated on page 3, the increase in NADCs in treated HIV-infected persons is mainly due to a prolonged life span! Perhaps one cannot formally exclude a cancer-inducing role of the antiviral drugs either (especially the nucleoside analogues), but I think there is no evidence for a direct role of HIV-1 or one of its protein/RNA products.

Authors' Response

NCOMM-17-01488B

Title: Exosomes derived from HIV-1-infected cells promote growth and progression of cancer via HIV TAR RNA.

Response to reviewers' comments:

We thank reviewers for their constructive criticisms and comments. We have added further experimental data and analysis and substantially revised the manuscript to address each of reviewers' comments. Following is our response to the comments.

Reviewer #1, Expertise: HIV related cancers/ virus related cancers

(Remarks to the Author):

Reviewer #1's comments: This manuscript has been improved since the original submission, and a number of the initial reviewers' comments addressed. The experiments on tumor samples and lymphomas are helpful in clarifying this phenomenon. As I indicated before, the potential role of exosomes from HIV-infected cells playing a role in tumor development in HIV patients is intriguing. However, some lingering concerns remain.

1. The question remains as to whether the concentration of the exosomes in the patients' plasma is really high enough to yield the tumor effects reported. The new data on tumor biopsies is somewhat helpful, but it is again unclear just how much TAR was present (this was RT-PCR which can be very sensitive). It is hard for me to tease this out -- in part, this is because the Figure legends often do not say much about the concentrations of exosomes used, and this should be spelled out so the reader can readily appreciate it. Also, in the response to the comment #1, the authors note that they typically obtained $15\text{-}20 \times 10^9$ exosomes per ml of patient plasma and that the results in Figure 2 (I assume 2a) were obtained with 4×10^9 exosomes per ml. If this is the case, then some of the key observations noted should be obtainable with unmodified patient plasma, and the effects potentially inhibited if exosome-depleted plasma is used. If this is the case, the paper would be strengthened by showing it. If not, or these calculations are incorrect, the authors should make it much clearer to the reader just how the effects seen relate to the concentrations in the patients' plasma and report the concentrations used throughout. In a similar way, it should be made clearer just how the animal experiments in Figure 2d were done.

Authors' response:

1) We did find that unmodified plasma from HIV+ patients enhanced cancer cell proliferation compared to that from healthy people, suggesting that the concentration of the exosomes in the patients' unmodified plasma is high enough to yield the tumor effect. The effect was inhibited when exosome-depleted plasma was used (now reported in Fig. 4d). We treated HSC3 cancer cells with unmodified plasma (equivalent to 4.7×10^9 exosomes/ml), exosomes purified from the plasma in serum-free medium (4.7×10^9 /ml) and resulting exosome-depleted plasma from the same HIV+ patients (n=6) and healthy individuals (n=6). The unmodified plasma of the HIV+ patients stimulated cell proliferation relative to that from healthy individuals. Exosome-depleted plasma from the patients, however, was unable to enhance cell proliferation. Exosome concentration in unmodified plasma and purified plasma were quantified and listed in Supplementary Table 3.

2) We reported qRT-PCR of the TAR RNA in tumor biopsies as relative to that in HIV+ 8E5/LAV T-cell exosomes. Each 8E5/LAV cell contains a single copy of HIV viral genome. Therefore, fold induction of TAR RNA in tumor biopsy over that in 8E5/LAV exosomes presented copy numbers of TAR. Since total RNA extracted from tumor FFPE biopsies contained RNA from not only exosomes but also all types of cells, we did not use “copy numbers” to quantify TAR RNA in the biopsies instead of relative fold induction.

3) We now report the concentrations of exosomes in all relevant results in figure legends. We also described the animal experiments in Figure 2d in more details.

Reviewer #1's comments:

a. In this regard, the authors say that the concentrations used is that used by previous authors (line 105), but it is unclear that the previous authors proved that these concentrations are biologically realistic. This is not, in itself, a sufficient argument.

Authors' Response:

We chose concentrations of exosomes mainly on our own experiments, including the results obtained using the unmodified plasma and the dose-response experiment (supplementary figures 2c). We removed the comparison of concentrations used in our report vs. previous authors since we reported different biological outcomes than others' previous work.

Reviewer #1's comments:

2. A number of the experiments (see Fig 3) describe effects on DEFB103, and this is used as a surrogate for the tumor promoting effect. However, it is not clearly explained why this is a good surrogate. This should be spelled out, or else other endpoints directly related to tumor development used.

Authors' Response:

We now use proto-oncogenes FOS and c-Myc as endpoints directly related to tumorigenesis. We found that exosomes from HIV+ T cells induced expression of proto-oncogenes FOS and c-Myc in HSC3 cancer cells. While the R06 aptamer blocked TAR-induced FOS expression, the mutant TAR RNA failed to stimulate its expression. We described the effects on c-Myc and FOS as relevant to cancer cell proliferation induced by TAR RNA-bearing exosomes (line 176). The results have been reported in Fig. 3i and j.

Reviewer #1's comments:

3. In the initial review, concern was raised that in Figure 3j, the scrambled R06 aptamer had some effect on suppressing DEFB103 ($p < 0.08$). The authors note they have changed the statistical test used. I cannot go back and see the original figure, but it appears that this control has now just been removed. If so, this is concerning and the control should be shown.

Authors' Response:

We are sorry for not addressing the concern properly. We modified the protocol for transfection of the R06 aptamer into exosomes. The results show that R06 transfection significantly blocked TAR RNA-bearing exosome-induced gene expression to the level of Jurkat cell exosomes did. Exosomes transfected with the scrambled RNA could still stimulate gene expression. The new results have been reported in Figure 3k with the control shown.

Reviewer #1's comments:

4. I have a number of relatively minor points in how the article is written that should be addressed:
 - a. In first 2 sentences of abstract, HIV is believed to play a key role in CERTAIN (but not all NADC). Also, not all mechanisms are obscure -- some are well known.
 - b. It is not really appropriate to generalize the results here to all NADC. Only 2 cancers in which this effect is seen are shown. I would either say "certain" NADC, or else focus all the description in the paper on the cancers studied.
 - c. Line 46. The incidence of cervical cancer was not dramatically reduced by ART.
 - d. On line 79 and throughout the manuscript, the authors use the term "progression". This term is imprecise. The authors should instead use descriptions of the actual effects being studied (migration, etc.)
 - e. On line 293, the authors speculate that exosomes could be used as a "pathogenic biomarker" for cancer progression, but to do so, they would have to show that the concentration of such exosomes differed greatly among patients and was predictive. I would tone this down.

Authors' Response:

- a. We now indicate that mechanisms are obscure in certain NADCs.
- b. We now focus all the description in the manuscript on the cancers studied.
- c. We removed cervical cancer and modified the description in line 44.
- d. We now use the description of the actual effects being studied, including proliferation, migration and invasion throughout the manuscript.
- e. We removed the description.

Reviewer #2, Expertise: exosomes (Remarks to the Author):

Reviewer #2's comments:

The authors have addressed many concerns a few are remaining.

The authors show 'relative TAR levels', while the fold increases may seem physiologically relevant at a first glance fig 4d-f, I wonder what the levels would be compared to infected HIV cells? i.e. could the authors correlate their TAR levels in exosomes and the tumor tissues to 'infected cell equivalents'? The reason I ask is whether the authors can exclude that a single HIV-infected T cell infiltrating the tumor mass could be responsible for the signals observed.

Authors' Response:

This is a good point! A group of HIV-infected infiltrating cells or a single cell may be able to affect tumor growth locally. The "relative TAR levels" was calculated based on TAR levels of exosomes from 8E5/LAV cells, which contain a single copy of HIV genome. In HIV+ HNSCC patient tumor tissues, higher CD4+ counts seemed to correlate with higher TAR levels in FFPE tissues (Fig. 4g vs. supplementary table 3). Nevertheless, this speculation was based on very limited numbers of patient biospecimens.

Reviewer #2's comments:

I still wonder if the effects seen are HIV-TAR exosome 'specific'. Another natural TLR3 ligand is the human tumor virus related EBER1 molecule, also found in exosomes (Iwakiri et al., 2009 JEM, Baglio et al PNAS 2016). Would exosomes with EBERs have similar effects? Could the authors transfect wild-type Jurkat exosomes with EBER1 and show similar (in vitro results) on cancer cell proliferation or perhaps

use LCL exosomes directly, assuming they also enter recipient cells via EGFR. This to me would provide more concrete evidence that exosomal small RNAs (from virus-infected cells) can have pro-tumorigenic effects mediated via TLR3.

Authors' Response:

We isolated exosomes from culture supernatants of EBV+ LCL lines NC-37 and IB4 (used by Baglio et al PNAS 2016) and two EBV- B cell lymphoma cell lines. Exosomes from EBV+ cells contained EBER1. Exosomes from EBV+ LCL cells could not enhance proliferation of cancer cells. In addition, neither EBV+ nor EBV- cells were able to stimulate expression of proto-oncogenes FOS and c-Myc. However, all four EBV+ and EBV- cells induced expression of the ISG IFIT1. These results have been reported in Supplementary Figure 6a, b and c. Our results suggest that the pro-tumor effect is HIV-TAR exosome "specific" in cancer cells studied.

Reviewer #2's comments:

Finally the in vivo data with the TLR3KO cells are compelling, however did the authors rule out a potential growth disadvantage of the KO cells?

Authors' Response:

TLR3-KO cells did not present growth disadvantage in our experiments (Fig. 6d).

Reviewer #3, Expertise: HIV/TAR infection (Remarks to the Author):

Reviewer #3's comments:

I still have a very major problem with the first half of the first sentence of the abstract. How do we know that HIV pathogenesis is contributing to NADCs? As correctly stated on page 3, the increase in NADCs in treated HIV-infected persons is mainly due to a prolonged life span! Perhaps one cannot formally exclude a cancer-inducing role of the antiviral drugs either (especially the nucleoside analogues), but I think there is no evidence for a direct role of HIV-1 or one of its protein/RNA products.

Authors' Response:

We now describe prolonged life/aging as a risk factor in NADCs (abstract) and claim that underlying mechanisms in certain NADCs are obscure.

In regarding evidence for a direct role of HIV-1 or one of its protein/RNA products: While the reviewer is definitely entitled to have the opinion, we have presented extensive data to support a role of exosomes derived from HIV-infected T cells (extensive in vitro and some in vivo data) and those from HIV+ patients (total 15 patient specimens) and TAR RNA (mutations and aptamers, proto-oncogene expression) in proliferation, migration and invasion of head neck carcinoma and lung cancer cells.

Reviewers' Comments:

Reviewer #1:

Remarks to the Author:

The paper is substantially improved. I find that the experiments with plasma from HIV patients contributes substantially to the argument that the in vitro results seen here have biological significance.

I do have one quibble regarding the revised first sentence of the abstract. While aging of the HIV population can in part explain the increasing incidence of certain NADC (compared to, say, 20 years ago), aging does not explain why these cancers are increased in the first place compared to HIV-uninfected persons. I would perhaps just say something like "People living with HIV/AIDS have an increased risk of non-AIDS-defining cancers (NADC), and the incidence of certain of these cancers is increasing as they live longer on anti-retroviral therapy." (If short on space, you could even leave out the second half of the sentence.)

Reviewer #2:

Remarks to the Author:

The authors have adequately addressed my points and I commend them with their efforts, this is a beautiful paper that enhance our understanding of virus-host interaction via exosomes.

Authors' Response

NCOMMS-17-01488C

Title: Exosomes derived from HIV-1-infected cells promote growth and progression of cancer via HIV TAR RNA.

Response to referees' comments:

We thank reviewers for their construction comments. We have revised the manuscript to address reviewers' comments. Following is our response to the comments.

Reviewer #1, Expertise: HIV related cancers/ virus related cancers (Remarks to the Author):

The paper is substantially improved. I find that the experiments with plasma from HIV patients contribute substantially to the argument that the in vitro results seen here have biological significance.

I do have one quibble regarding the revised first sentence of the abstract. While aging of the HIV population can in part explain the increasing incidence of certain NADC (compared to, say, 20 years ago), aging does not explain why these cancers are increased in the first place compared to HIV-uninfected persons. I would perhaps just say something like "People living with HIV/AIDS have an increased risk of non-AIDS-defining cancers (NADC), and the incidence of certain of these cancers is increasing as they live longer on anti-retroviral therapy." (If short on space, you could even leave out the second half of the sentence.)

Authors' response:

We thank the reviewer for the comments on the revision and the effort. We now start the abstract with the sentence "People living with HIV/AIDS on antiretroviral therapy have an increased risk of non-AIDS-defining cancers (NADC)". We have to leave out the second half of the sentence because limits in space.

Reviewer #2, Expertise: exosomes (Remarks to the Author):

The authors have adequately addressed my points and I commend them with their efforts, this is a beautiful paper that enhances our understanding of virus-host interaction via exosomes.

Authors' response:

We greatly appreciate the reviewer's comments that helped improve the manuscript. We thank the reviewer for the effort.